



# Reconstructing ice phenology of lake with complex surface cover: A case study of Lake Ulansu during 1941–2023

Puzhen Huo[1], Peng Lu[1], Bin Cheng[2], Miao Yu[1], Qingkai Wang[1], Xuewei Li[1], Zhijun Li[1]

[1]State Key Laboratory of Coastal and Offshore Engineering, Dalian University of Technology, Dalian, 116024, China

[2]Finnish Meteorological Institute, Helsinki, 00101, Finland

*Correspondence to*: Peng Lu (lupeng@dlut.edu.cn)

**Abstract.** Lake ice phenology plays a critical role in determining the hydrological and biogeochemical dynamics of the catchment and regional climate. Lakes with complex shorelines and abundant aquatic vegetation are challenging for lake ice phenology retrieval using remote sensing data, primarily due to mixed pixels containing plants, land and ice. To tackle this

challenge, a new double-threshold moving *t* test (DMTT) algorithm, utilizing multisource satellite-derived brightness temperature data at a 3.125-km resolution and long-term weather data, was introduced to capture Lake Ulansu's ice phenology from 1979 to 2023. Compared to the previous moving *t* test algorithm, the new DMTT algorithm employs air temperature time series to assist in determining abrupt change points and uses two distinct thresholds to calculate the freeze-up start (FUS) and break-up end (BUE) dates. This method improved the detection of ice information effectively for the

mixed pixels. Furthermore, we extended Lake Ulansu's ice phenology detection backward to 1941 using a random forest (RF) model. The reconstructed ice phenology from 1941 to 2023 indicated that Lake Ulansu had average FUS and BUE dates of November 15 ± 5 and March 25 ± 6, respectively, with an average ice cover duration of 130 ± 8 days. Air temperature was the primary impact factor, accounting for 56.5 % and 67.3 % of the variations in the FUS and BUE dates, respectively. We reconstructed, for the first time, the longest ice phenology over a large shallow lake with complex surface cover. We argue

DMTT can effectively be applied to retrieve lake ice phenology for this type of lake that have not been fully explored worldwide.

## 1 Introduction

Global warming compels us to examine the impact of climate on the environment. One striking fact is the reduction in the ice component in the cryosphere (Kang et al., 2012; Li et al., 2022). Lake ice, plays a vital role in the Earth's climate system

and ecological balance (Latifovic and Pouliot, 2007; Wu et al., 2022). When lakes freeze, the ice cover not only alters their physical state but also affects the exchange of heat, momentum, and mass between the atmosphere and the water body, profoundly influencing the local climate and environment (Aslamov et al., 2014). Ice phenology data provides valuable insights into understanding and predicting climate change (Mishra et al., 2011; Woolway et al., 2020). Benson et al. (2011) analyzed lake ice phenology over a century in the Northern Hemisphere. However, the long-term ice phenology trends,

especially for shallow lakes with complex shorelines and abundant aquatic vegetation, which dominate the Northern Hemisphere (Pi et al., 2022), remain largely unexplored due to a lack of historical observational data. These lakes are more



sensitive to climate change than deep lakes because of their low heat capacity (Ambrosetti and Barbanti, 1999), potential ecological effects on algal blooms (Duan et al., 2012) and shortened ice cover duration. Studying ice phenology in unique lake ecosystems, such as these, not only reveals the complex microlevel interactions between climate, water, and biology
(Sharma et al., 2021; Smits et al., 2021), but also enhances our broader understanding of climate change impacts, which helps refine global climate models and improve weather forecasts. This focus fills a critical knowledge gap in how climate change affects diverse freshwater systems (Yang et al., 2020).

In situ observations provide the best quality information on ice phenology (Benson et al., 2011). However, due to environmental and logistical constraints, sustained manual monitoring is challenging (Jewson et al., 2009). Remote sensing
data, with their broader temporal and spatial monitoring capabilities, have been widely used in the study of lake ice phenology (Wang et al., 2021; Wu et al., 2022). For example, Moderate Resolution Imaging Spectroradiometer (MODIS) 8-day composite data have been used to determine ice phenology (Kropáček et al., 2013). Optical satellites offer data with higher spatiotemporal resolution, but they depend on solar energy reflected from the Earth's surface, making them susceptible to cloud cover and lighting conditions (Murfitt and Duguay, 2021). In contrast to optical remote sensing,
microwave remote sensing allows monitoring the Earth's surface under most weather conditions (Nunziata et al., 2021). Active microwave remote sensing is primarily utilized for extracting ice phenology data in large water bodies due to its lower temporal resolution (Antonova et al., 2016; Howell et al., 2009). This is because in these lakes, the phase transition between water and ice takes a longer time. Passive microwave remote sensing can provide data with longer temporal coverage but coarse spatial resolution, making them more suitable for large lake ice phenology detection (Kang et al., 2012;
Su et al., 2021). Additionally, a common challenge to this method are mixed pixels. These are pixels that contain multiple surface types, such as water, vegetation, and land, which are especially common around complex shorelines (Bellerby et al., 1998). To mitigate the impact of mixed pixels, buffer zones are typically implemented within passive microwave remote sensing technology, effectively shrinking the shoreline when extracting pixel information (Cai et al., 2022). The latest Calibrated Enhanced-Resolution Passive Microwave Daily EASE-Grid 2.0 offers brightness temperature data from 1979 to
the present, with a spatial resolution as high as 3.125 km. While this advancement significantly improves the monitoring capabilities of the Earth's surface (Johnson et al., 2020), it still poses challenges in accurately capturing data for small lakes or lakes with complex shorelines due to mixed pixels. In addition, if long-term lake ice phenology records are desired, satellite remote sensing can also provide training datasets for machine learning (Wu et al., 2021; Xu et al., 2024). The success of these models largely depends on the quality of the training data, which requires not only accuracy but also a
comprehensive temporal record. Ruan et al. (2020) showcased a practical machine-learning application by using a random forest model, which, by leveraging robust datasets and CMIP6 projections, forecasts the ice phenology of lakes on the Tibetan Plateau up to 2099. This shows the critical need for reliable and comprehensive ice phenology data, especially for lakes with complex surface characteristics, to fully utilize the potential of machine-learning approaches.

This study took Lake Ulansu as an example; this lake is the largest aquatic plant-dominated shallow lake in Northwest China
and is characterized by its rich aquatic vegetation and complex shoreline. The objective of this study was to develop an





automated algorithm that overcomes the mixed-pixel challenges posed by rich aquatic vegetation and complex shorelines, thus enabling an accurate classification of ice and water states and obtaining reliable ice phenology data for long-term reconstructions. Specifically, our research strategy was composed of the following steps: (1) A new algorithm was developed to classify ice and water states in brightness temperature data from satellite remote sensing for the period 1979–2023. (2) A random forest model was utilized with meteorological data to reconstruct the ice phenology from 1941 to 1978. (3) The meteorological impact on the ice phenology of Lake Ulansu from 1941 to 2023 was analyzed to explore the key drivers of its variations.

## 2 Study area

Lake Ulansu (40°46′–41°7′ N, 108°41′–108°58′ E) is situated on the eastern side of the Hetao Basin in North China (Fig. 1a). The lake covers an area of approximately 300 km$^2$. It has an elevation of approximately 1,050 m, a shoreline perimeter of approximately 140 km, and a north–south width ranging from 35 to 45 km, while its east–west width is narrower, at approximately 5 to 10 km. The water depth varies from 0.5 to 3 m, with an average depth of 1.6 m, indicating that the lake is a typical shallow lake. As an important water resource in the Hetao Basin, Lake Ulansu profoundly impacts local agriculture, irrigation, residents' livelihoods, and the regional climate (Guo et al., 2008; Li et al., 2020). The lake is rich in aquatic vegetation (Fig. 1c), which accounts for approximately 60 % of the lake's area (i.e., 180 km$^2$). The transpiration of reeds and lake evaporation result in considerable water loss, with an annual evaporation of approximately 2,300 mm. Precipitation in the lake area primarily occurs during the summer, with an annual precipitation of approximately 220 mm. The lake's water depth and volume are primarily regulated during the summer through water diversion from the Yellow River, with minimal surface water inflow or outflow occurring during the winter (Huang et al., 2022). From 1958 to 2015, the Hetao Basin had an annual average air temperature of approximately 7.0 ℃, with average air temperatures in January and July reaching −10.1 ℃ and 23.8 ℃, respectively (White et al., 2020). Typically, the freeze-up start (FUS) date occurred in November, and the break-up end (BUE) date occurred in March, with an average ice cover duration (ICD) of 127 days during 2013–2022 (Huo et al., 2022). Each year in this study is hereafter defined from August 1st to July 31st of the following year for simplicity.



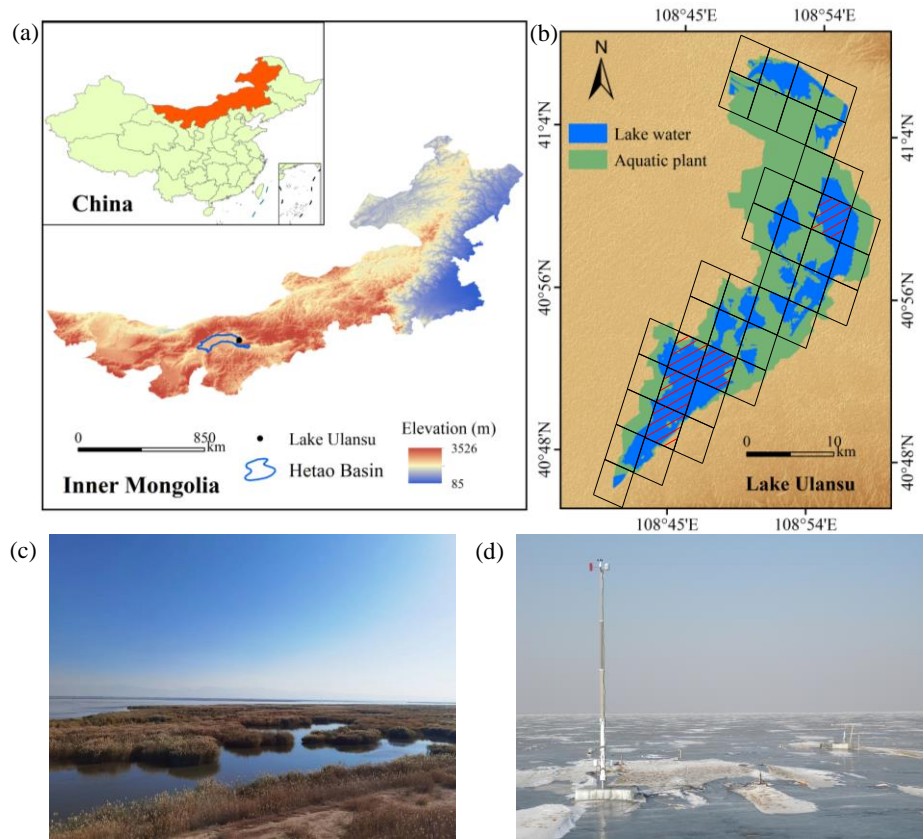


**Figure 1: (a) Location and geometry of Lake Ulansu. (b) CETB data grids with shaded areas representing brightness temperature pixels selected for Lake Ulansu surface identification. (c) Aquatic reeds within Lake Ulansu. (d) On-ice instrumentation for field observations (Lu et al., 2020).**

## 3 Data and methods

Three datasets and two algorithms were used of this study, as shown in the flowchart (Fig. 2). First, the double-threshold moving $t$ test algorithm and ERA5 air temperature data were used to detect abrupt change points in the passive microwave brightness temperature series from satellites. These points helped classify brightness temperature pixels as ice or water states, allowing the determination of the FUS and BUE dates for Lake Ulansu from 1979 to 2023. These dates were also validated

by ice phenology data obtained from satellite optical data from 2000 to 2023. Furthermore, meteorological and ice phenology data spanning from 1979 to 2023 were used to train a random forest model. Subsequently, we input the ERA5 data into the random forest model to obtain historical ice phenology data for Lake Ulansu from 1941 to 1978.



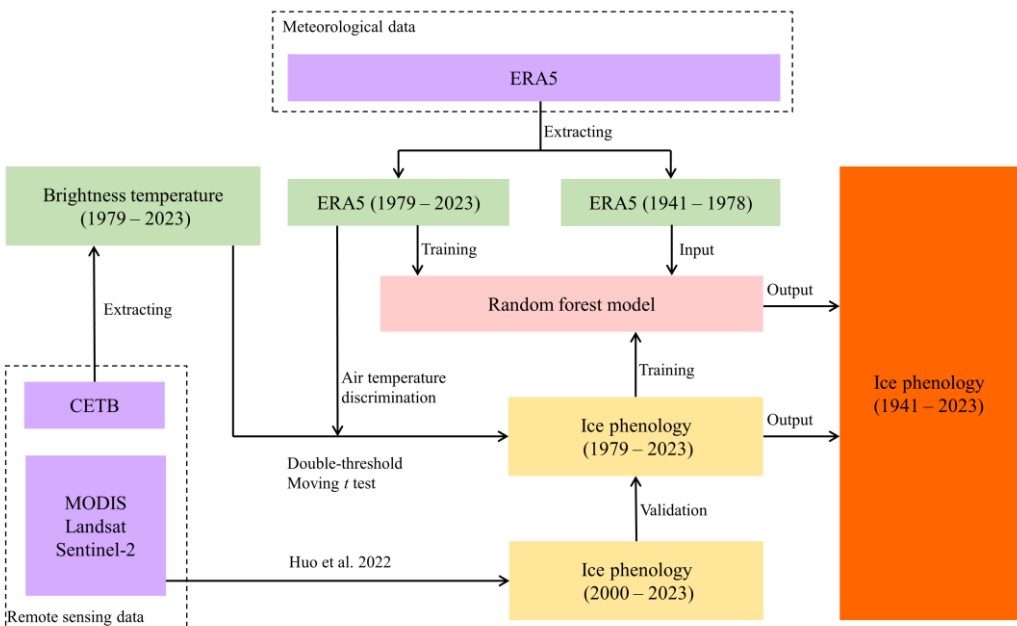

**Figure 2: Flowchart of the data and methodology used in this study.**


## 3.1 Data

### 3.1.1 Brightness temperature data

In this study, we used Calibrated Enhanced-Resolution Passive Microwave Daily EASE-Grid 2.0 Brightness Temperature (CETB) data at 37 GHz H-polarization with a high spatial resolution of 3.125 km provided by the National Snow and Ice

Data Center (NSIDC). The CETB data include an abundance of microwave radiation brightness temperature ($T_b$) data, facilitating the investigation of Earth's surface characteristics. The CETB data were obtained from various satellite sensors, including SMMR on Nimbus 7, SSM/I and SSMIS on the DMSP satellite series, and AMSR-E on Aqua (Brodzik et al., 2016). CETB data boast high spatial resolution and comprehensive record features. The microwave radiation $T_b$ data were collected from various channels and time intervals, and were stored with grid spacings ranging from 3.125 km to 25 km. The

CETB grid employed the "drop-in-the-bucket" average (GRD) algorithm to map essential location data onto output grid units, resulting in $T_b$ data with a 25-km resolution. Subsequently, the resolution was enhanced using the radiometer version of the scatterometer image reconstruction (rSIR) algorithm, which yielded $T_b$ data with a resolution of 3.125 km to 12.5 km. Due to the narrow and irregular shoreline of Lake Ulansu, high spatial resolution $T_b$ data are essential. Furthermore, the CETB data exhibited a relatively coarse spatial resolution of 6.25 to 12.5 km at frequencies below 30 GHz (Brodzik et al., 2016), with

higher frequencies (near 90 GHz) being more susceptible to weather effects (Ivanova et al., 2015). Considering these factors and referencing the study by Cai et al. (2022) regarding the efficient coverage of $T_b$ pixels by various sensors within the



CETB data, $T_b$ data with a resolution of 3.125 km were used in this study. These data were sourced from the SMMR on the Nimbus 7 satellite; the SSM/I on the F08, F10, F11, F13, F14 and F16 satellites; and the SSMIS on the F16 satellite.

### 3.1.2 Optical satellite data

In this study, ice phenology data for Lake Ulansu from 2000 to 2023 were compiled (Huo et al., 2022). These data originated from cloud information from the state_1 km_1 parameter of the MOD09GA and MYD09GA datasets, effectively reducing the influence of clouds on optical sensors by using the spatiotemporal features of MODIS datasets to replace cloud-covered pixels. Additionally, single-band thresholding and dynamic thresholding methods were employed to classify the reflectance band of the MOD09GQ and MYD09GQ datasets (Zhang and Pavelsky, 2019). This enabled the iterative extraction of water

and ice pixels and their comparison with higher spatial resolution Landsat and Sentinel-2 datasets. The goal was to identify optimal thresholds for distinguishing the proportion of ice and water coverage and to obtain a long-term time series of these data. By employing thresholds of 20 % and 80 % to differentiate the proportion of ice and water pixels in the total pixel count, it was possible to calculate the FUS and BUE dates. The ICD was then derived from the difference between these two dates. Due to the higher spatiotemporal resolution of the MODIS products, these three ice phenology data points agreed with

field observations (Huo et al., 2022). In this study, ice phenology data obtained from optical satellites were used to validate the ice phenology results obtained using the $T_b$ data.

### 3.1.3 Meteorological data

     In this study, the ERA5 meteorological dataset, which was provided by the European Centre for Medium-Range Weather Forecasts (ECMWF), was utilized for comprehensive analysis. ERA5 is the latest reanalysis dataset that integrates model

data with global observational data to produce a consistent dataset with a spatial resolution of 0.25° × 0.25° and a temporal resolution of 1 hour. In this study, a collection of ERA5 hourly data on single levels from 1940 to the present was chosen. The selected meteorological variables included air temperature, wind speed, incident solar radiation and precipitation (including rainfall and snowfall), spanning from 1941 to 2023.

     Although meteorological stations are sparse in the Hetao Basin, a comprehensive evaluation of the suitability of ERA5 data

for Lake Ulansu is still necessary. The daily average air temperature, wind speed, and incident solar radiation were calculated and compared with available field observations from 2016 to 2018 and from 2022 to 2023 (Cao et al., 2021; Lu et al., 2020). As shown in Fig. 3a, there was a significant agreement between the ERA5 air temperature and field observations, with a correlation coefficient ($R$) of 0.99. The mean absolute error ($MAE$) was 0.95 °C, and the root mean square error ($RMSE$) was 1.17 °C. Further error reduction was achieved by applying a regression equation to adjust the air temperature.

This high accuracy of air temperature ensures the precision of subsequent ice phenology calculations based on CETB data. Although the reanalysis gridded data may struggle to capture near-surface true wind speeds, the ERA5 wind speed data exhibited the greatest consistency in inland areas compared to other reanalysis datasets (Ramon et al., 2019). A correlation





coefficient of $R = 0.81$ was achieved by comparison with field observations, as shown in Fig. 3b, and the error for wind speed data over Lake Ulansu further decreased to a $MAE = 0.48$ m s$^{-1}$ and $RMSE = 0.61$ m s$^{-1}$ after linear regression was
applied. Overall, the incident solar radiation data from this dataset was slightly lower than that from the field observations (Fig. 3c), but the corrected daily averages were within acceptable limits, with a $MAE$ of 18.04 W m$^{-2}$ and a $RMSE$ of 22.35 W m$^{-2}$. Previous studies have indicated that using ERA5 precipitation data has significant advantages and they are widely used in various regions (Bandhauer et al., 2021; Yuan et al., 2021). In this study, cumulative precipitation was calculated for the ice phenology trend analyses for Lake Ulansu. Its validations are not presented here because of the absence of in-situ
precipitation observations (Fig. 1d) in this region with only limited snowfall or thin snow cover (< 4 cm) occurring in the winter (Lu et al., 2020).

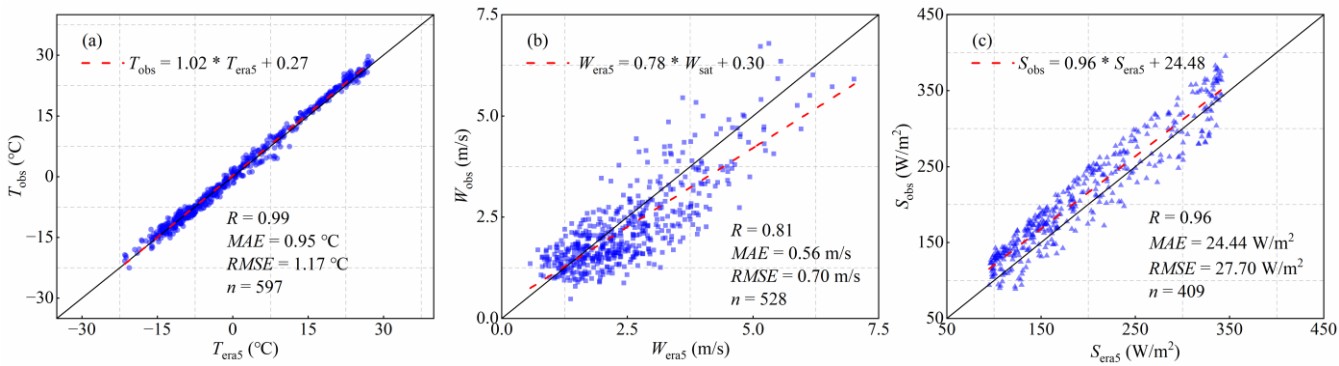

**Figure 3: Comparison between ERA5 and observed daily average air temperature (a), wind speed (b), and incident**
**solar radiation (c) meteorological data for Lake Ulansu. *R*, *MAE*, *RMSE*, and *n* denote the correlation coefficient, mean absolute error, root mean square error, and sample size, respectively.**

## 3.2 Methods

### 3.2.1 Identification of lake surface states

In this study, the ice phenology from 1979 to 2023 was determined by the brightness temperature ($T_b$) of the lake surface because $T_b$ changes considerably when a phase change occurs on the lake water surface (Su et al., 2021), providing a clear reference for determining the onset and end of the ice period. However, due to the complex shape of the Lake Ulansu shoreline, which is characterized by its narrow form, most grid cells encompass not only water but also aquatic vegetation and land. As a result, five $T_b$ grids with a proportion of lake water greater than 0.70 were selected to represent the status of
the lake surface (Fig. 1b). Figure 4a illustrates the annual variation in $T_b$ for each grid of each year as blue dashed lines, with the blue solid line showing the average for all grids during 1979–2023 and a solid red line depicting the average air temperature. The annual variation in the $T_b$ of Lake Ulansu exhibited a typical 'W' shape (Fig. 4a). This is different from



what is seen for large lakes with only water surfaces where $T_b$ is typically a pure pixel, resulting in a line with an 'Ω' shape (Cai et al., 2022; Su et al., 2021). Thus, previous methods to address the Ω-shaped $T_b$ series are not available in this study, and a double-threshold moving $t$ test (DMTT) algorithm was developed to extract the ice phenology from the W-shaped $T_b$ series.

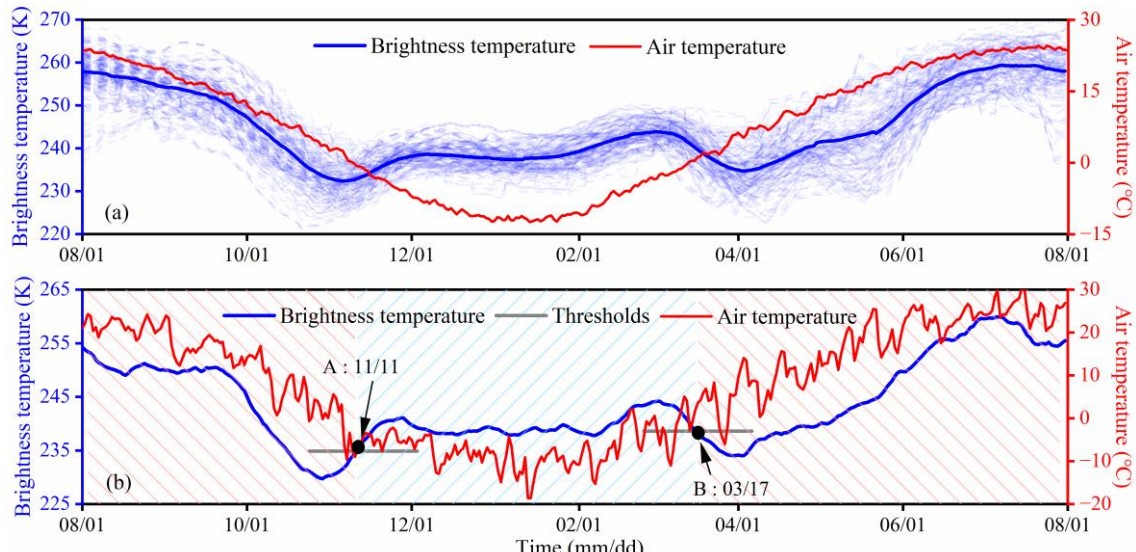

**Figure 4: Time series of brightness temperature (blue dashed lines) between 1 August and 31 July from 1979 to 2023 for Lake Ulansu. The solid lines represent the interannual average brightness temperature and air temperature time series, respectively. (b) The brightness temperature (blue solid line) and air temperature (red solid line) for the year 2001. The ice and water statuses were determined using the double-threshold moving $t$ test algorithm, where the red and blue shaded regions represent the water and ice states, respectively.**

The DMTT originated from the moving $t$ test (MTT), which was initially employed to identify abrupt meteorological change points in time series (Jiang and You, 1996; Shi and Zhu, 1996) and was later adapted by Du et al. (2017) to detect the abrupt change point (ACP) of $T_b$ in passive microwave time series, enabling the discrimination of lake ice phenology. In this study, we enhanced the MTT to identify the ACP of the $T_b$ series of complex mixed pixels in Lake Ulansu. Daily air temperature was also introduced to identify more suitable ACPs, facilitating the computation of the FUS and BUE dates. For the detailed steps and formulas of the MTT, refer to Du et al. (2017). Here, we specifically highlight the improvements of the DMTT:

Step 1: In addition to adhering to the ACP criteria outlined by Du et al. (2017), we conducted separate ACP detection for $T_b$ from August to December and from January to July. Furthermore, ACPs in the period from August to December were required to have air temperatures below 0 °C, while ACPs in the period from January to July were required to have air temperatures above 0 °C. The purpose of these two enhancements was to accommodate the variations in mixed pixels during the transitions between water and ice.





Step 2: After detecting multiple ACPs in Step 1, we calculated $\overline{T_{b1}}$ and $\overline{T_{b2}}$ for each ACP, which represented the mean $T_b$ 20 days before and after the ACP, respectively. The freezing threshold was determined as the mean of the minimum $\overline{T_{b1}}$ and its corresponding $\overline{T_{b2}}$ within the same group, while the melting threshold was defined as the mean of the minimum $\overline{T_{b2}}$ and its associated $\overline{T_{b1}}$ within the group. The DMTT algorithm calculated distinct thresholds for FUS and BUE dates (gray solid lines

in Fig. 4b) which made it possible to handle various ice/water transition scenarios. This adaptability to different $T_b$ series ensured the robustness of the algorithm (Appendix A).

Step 3: We classified each $T_b$ series using two thresholds to determine the ice and water status of the pixels. As demonstrated in Fig. 4b for a grid in Lake Ulansu in 2021, the interface between the red and blue shaded regions indicates the transitions between states, with points A and B marking the dates of these transitions in the $T_b$ series. Given that 5 CETB grids (Fig. 1b)

were selected within Lake Ulansu, the first date in the year when the status transitioned from water to ice was defined as the FUS date. Similarly, the first date when the transition occurred from ice to water was defined as the BUE date, with the ICD being the difference between these two. Based on this, we applied the DMTT algorithm to calculate thresholds and classify the CETB data, obtaining the ice phenology of Lake Ulansu from 1979 to 2023.

The results of the DMTT algorithm were validated through comparison with ice phenology data obtained from multisource

optical satellite data covering the period from 2000 to 2023. Figure 5 reveals a remarkably high correlation for the FUS dates ($R = 0.92$), with a minimal *MAE* of 2.00 days and a *RMSE* of 2.56 days. The correlation for the BUE date is slightly lower ($R = 0.87$), with a somewhat greater *MAE* of 2.67 days and a *RMSE* of 3.25 days. However, the *MAE* and *RMSE* for the ICD calculated based on the difference between the BUE and FUS dates remain within 5 days, indicating no systematic bias but random errors. Additional comparisons were also conducted using both the MTT algorithm and limited field observations

(Appendix B), and the DMTT algorithm still demonstrated better adaptation than the other algorithms to the scenario for Lake Ulansu, which is characterized by complex shorelines and abundant aquatic vegetation.

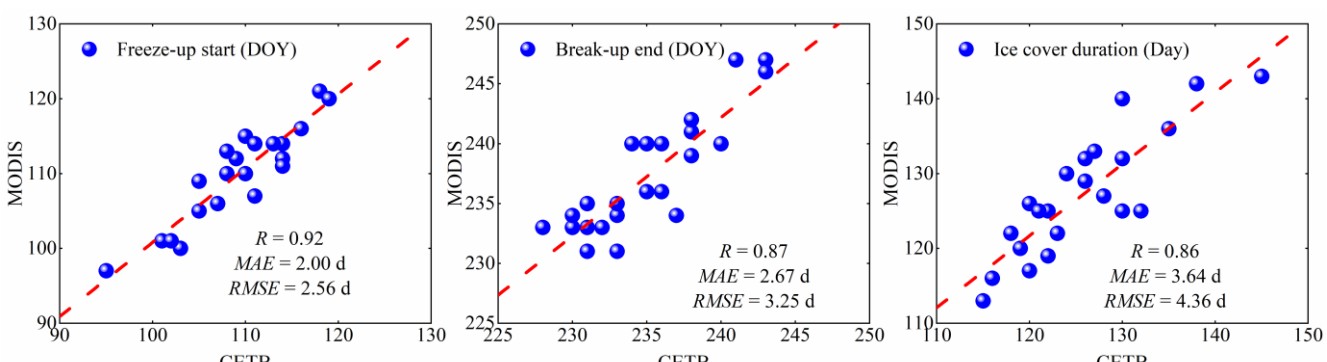

**Figure 5: Comparison of the ice phenology extracted by the CETB data and from multisource optical satellites (2000–**
**2023) for Lake Ulansu (The red dashed line represents the linear regression).**





### 3.2.2 Estimation of ice phenology with the random forest model

For the period from 1941 to 1978, when satellite remote sensing data were unavailable, the random forest (RF) algorithm was employed to estimate the ice phenology of Lake Ulansu. The RF model is a powerful ensemble learning technique widely used for data modeling and prediction. The model accuracy can be improved by integrating the results of multiple decision trees (Breiman, 2001). Model robustness is particularly crucial when dealing with complex meteorological data for predicting ice phenology (Ruan et al., 2020). Previous research has highlighted the influence of short-term meteorological factors on the freezing and melting of shallow lakes (Blagrave and Sharma, 2023; Caldwell et al., 2020). Hence, we opted to employ a three-month meteorological dataset to train the RF model, aligning with the characteristics of Lake Ulansu.

To train and validate the RF model, we randomly divided the dataset, which includes meteorological factors and ice phenology from 1979 to 2023, into two subsets. Seventy percent of the data were designated for training purposes, while the remaining 30 % were set aside for validation. Each subset was further segmented into two parts: the first part comprised the average air temperature, wind speed, solar radiation, and cumulative precipitation for September, October, and November, which were utilized in predicting the FUS dates. The second part mirrored the first in terms of meteorological variables but focused on the months of January, February, and March to predict the BUE dates. This organization resulted in each segment contributing twelve data points toward the prediction of each target variable, encapsulating the critical meteorological factors over the respective three-month periods.

We employed the same hyperparameters (number of trees = 10, 20, 50, 100) as Anilkumar et al. (2023) did for glacier mass balance studies. Utilizing grid search and evaluation, we considered the $R$, $MAE$, and $RMSE$ to select the hyperparameters. The optimal configuration was identified as 20 trees for both the FUS and BUE models (Appendix C), and 3-fold cross-validation was conducted on the training set. To further evaluate the efficacy of the RF model, we employed the $R$, $MAE$, and $RMSE$ to assess its performance on the validation set. As shown in Table 1, the RF model outperformed the BUE model ($R = 0.94$) on the FUS model ($R = 0.96$) within the training set. The $MAEs$ for both models were less than 2 days. Furthermore, it is also worth mentioning the results from the validation set deserve, with both models achieving $R$ values exceeding 0.8, underscoring the precision of the RF model in predicting the ice phenology of Lake Ulansu.

**Table 1. Performance of the random forest model in predicting the freeze-up start date and the break-up end date. ($R^2$, $MAE$, and $RMSE$ denote the coefficient of determination, mean absolute error, and root mean square error, respectively)**

| Ice phonology | Training | | | Validation | | |
|---|---|---|---|---|---|---|
| | $R$ | $MAE$ (d) | $RMSE$ (d) | $R$ | $MAE$ (d) | $RMSE$ (d) |
| Freeze-up start | 0.96 | 1.35 | 1.87 | 0.94 | 3.21 | 3.85 |
| Break-up end | 0.94 | 1.55 | 2.02 | 0.80 | 4.74 | 5.37 |



After the RF model was established and validated, we used ERA5 meteorological data from 1941 to 1978, including data from September to November and from January to March of each year, as input features to obtain the historical ice phenology of Lake Ulansu.

## 4 Results

### 4.1 Ice phenology during 1941-2023

The ice phenology, including the FUS and BUE dates and ICD of Lake Ulansu from 1941 to 2023, is shown in Fig. 6. It is clear that the ice phenology trends after 1982 were somewhat opposite to those before 1982, with this reversal being particularly pronounced in the ICD. To provide a detailed depiction of this reversal, we divided the whole period into two subperiods in the following analyses, and the corresponding statistical results are presented in Table 2.

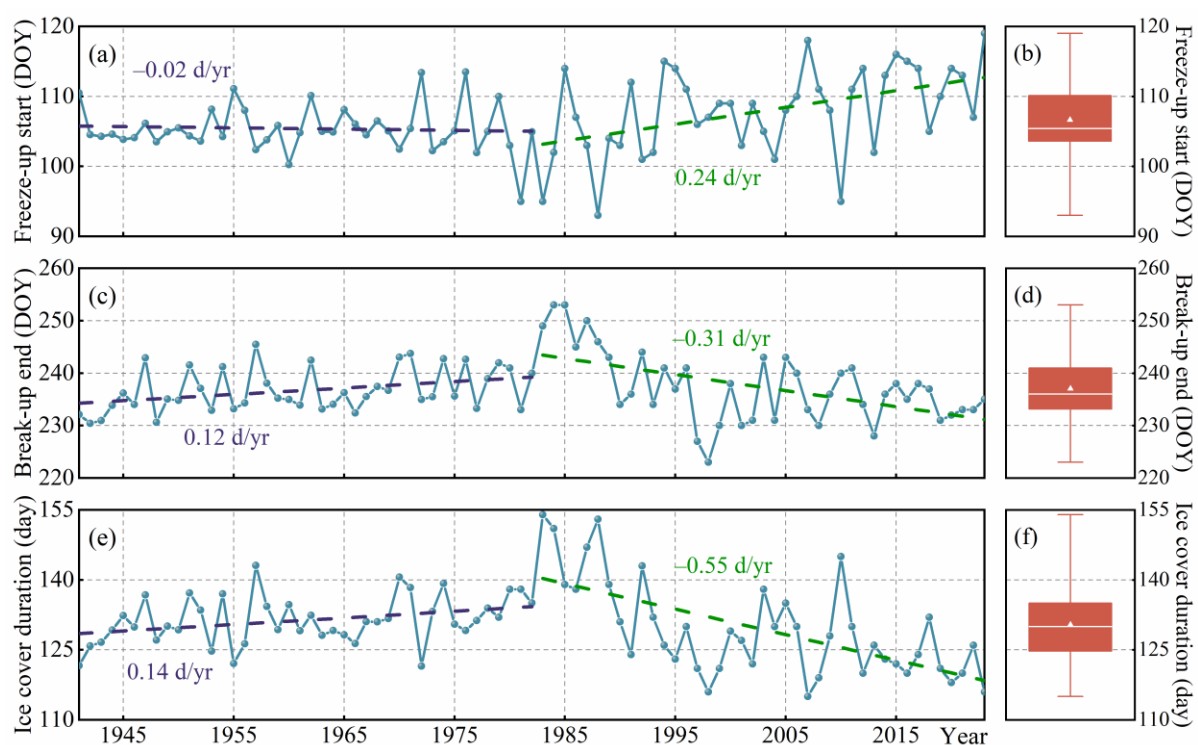


**Figure 6: Ice phenology results for Lake Ulansu from 1941 to 2023. (a) and (b): Freeze-up start dates. (c) and (d): Break-up end dates. (e) and (f): Ice cover duration. The solid lines indicate annual data, and the dashed lines represent linear trends for 1941–1982 and 1983–2023. Box plots depict the distribution of ice phenology data.**

**Table 2. Averages (Avg) and linear trends (Lt) of ice phenology in Lake Ulansu for various periods covering the years 1941 to 2023. Significance levels for trends are denoted by * (p < 0.05) and ** (p < 0.01).**



| Period | Freeze-up start | | Break-up end | | Ice cover duration | |
|---|---|---|---|---|---|---|
| | Avg (DOY) | Lt (d yr$^{-1}$) | Avg (DOY) | Lt (d yr$^{-1}$) | Avg (d) | Lt (d yr$^{-1}$) |
| 1941–1982 | 105.34 | −0.02 | 236.75 | 0.12* | 131.41 | 0.14* |
| 1983–2023 | 108.00 | 0.24** | 237.37 | −0.31** | 129.37 | −0.55** |
| 1941–2023 | 106.65 | 0.07** | 237.05 | −0.01 | 130.40 | −0.08* |

During the period from 1941 to 1982, the average date for FUS was approximately the 105th day (November 13), with an advance of 0.02 d yr$^{-1}$. The BUE date trend was more significant, with a delay of 0.12 d yr$^{-1}$, resulting in an average increase of 0.14 d yr$^{-1}$ in the ICD. Notably, the three ice phenology trends from 1983 to 2023 are highly significant ($p < 0.01$). During this period, the FUS date exhibited a delay trend, advancing by an average of 0.24 d yr$^{-1}$, while the BUE date showed an early trend, advancing by an average of 0.31 d yr$^{-1}$. However, the ICD decreased by an average of 0.55 d yr$^{-1}$. These trends contrast with the ice phenology trends from 1941 to 1982 and are characterized by more pronounced trends.

Overall, from 1941 to 2023, the ice phenology in Lake Ulansu exhibited several notable features. The FUS occurred between the 93rd and 119th days, with an average date of approximately the 107th day (November 15). The BUE ranged from the 223rd to the 237rd day, typically occurring around the 237th day (March 25). The ICD spanned from 115 to 154 days, with an average of approximately 130 days.

**4.2 Meteorological factor contributions to ice phenology**

Ice phenology is primarily determined by local meteorological conditions, and Table 3 summarizes the variations in the key meteorological factors influencing Lake Ulansu's ice phenology from 1941 to 2023. It is evident that the changes in air temperature are consistent with those in ice phenology, as shown in Fig. 6. During 1941–1982, the average air temperature decreased by 0.04 °C yr$^{-1}$ from September to November and accelerated to 0.05 °C yr$^{-1}$ from January to March. This cooling trend coincided with an advanced FUS date and a delayed BUE date, extending the ICD (Table 2). Conversely, in the period from 1983 to 2023, both periods experienced increases in average air temperature at rates of 0.04 °C yr$^{-1}$ and 0.08 °C yr$^{-1}$, respectively. This warming trend was associated with a delayed FUS date and an advanced BUE date, indicating a decreased ICD. The observed air temperature fluctuations echo broader climate changes, showing similar trends to those observed in Lake Mendota (Magee et al., 2016). Importantly, these shifts, which are particularly pronounced during the ice season, underscore the tight connection between ice phenology and regional climate dynamics (Marengo and Camargo, 2007).

However, variations in other meteorological factors are not as pronounced as those in air temperature. The wind speed remained relatively stable across these years (Table 3). The incident solar radiation decreased notably from September to November in accordance with the autumn-winter transition. The majority of precipitation occurred during the non-ice season. The months from September to November accounted for 18 % to 24 % of the annual precipitation, while the months from January to March contributed approximately 6 %, suggesting a minimal impact of snow cover formation on the ice surface due to its thinness.






**Table 3. Averages (Avg) and linear trends (Lt) of meteorological factors in Lake Ulansu for various periods covering the years 1941 to 2023. Significance levels for trends are denoted by * ($p < 0.05$) and ** ($p < 0.01$).**

| Meteorological factor | Period | 1941–1982 | | 1983–2023 | | 1941–2023 | |
|---|---|---|---|---|---|---|---|
| | | Avg | Lt | Avg | Lt | Avg | Lt |
| Air temperature (°C) | Annual | 6.80 | −0.04** | 7.41 | 0.05** | 7.10 | 0.01** |
| | Sept.–Nov. | 6.75 | −0.04** | 7.27 | 0.04** | 7.01 | 0.01* |
| | Jan.–Mar. | −6.60 | −0.05* | −5.78 | 0.08** | −6.19 | 0.02* |
| Wind speed (m s⁻¹) | Annual | 2.59 | 0.00 | 2.55 | 0.00* | 2.57 | 0.00 |
| | Sept.–Nov. | 2.39 | 0.00 | 2.39 | 0.00 | 2.39 | 0.00 |
| | Jan.–Mar. | 2.23 | 0.00 | 2.24 | 0.00 | 2.24 | 0.00 |
| Incident solar radiation (W m⁻²) | Annual | 226.45 | −0.03* | 224.90 | −0.02 | 225.68 | −0.03** |
| | Sept.–Nov. | 183.53 | −0.07** | 181.19 | −0.04** | 182.38 | −0.06** |
| | Jan.–Mar. | 171.90 | 0.01 | 171.54 | 0.02 | 171.72 | 0.00 |
| Precipitation (mm) | Annual | 350.84 | −4.56 | 269.55 | −1.75 | 310.69 | −2.27** |
| | Sept.–Nov. | 66.75 | 0.30 | 63.25 | −0.06 | 65.02 | −0.03 |
| | Jan.–Mar. | 20.18 | −0.05 | 17.28 | −0.32* | 18.75 | −0.10 |

To further understand the overarching influence of these meteorological factors, the previous RF model was also employed

to rank the importance of each factor and determine their cumulative contributions to the FUS and BUE dates over specific periods (9–11 months and 1–3 months). As the predominant driver of changes in ice phenology, air temperature accounted for 56.5 % and 67.3 % of the variation in the FUS and BUE dates, respectively (Fig. 7). This approach is straightforward because air temperature is the most important factor, among all meteorological factors, for controlling the heat balance at lake surfaces (Imrit and Sharma, 2021; Kropáček et al., 2013). Following air temperature, incident solar radiation

contributed 20.1 % to the variation in FUS dates and 15.0 % to the variation in BUE dates. As seasons transitioned, the decreasing intensity of solar radiation reduced its heating effect on lake ice. The influence of precipitation was slightly more pronounced on the BUE date (14.7 %) than on the FUS date (12.3 %). Despite the limited precipitation in winter in Lake Ulansu, the snow cover significantly altered the heat budget of the lake ice. The high albedo of snow cover reflected a considerable amount of solar radiation, while its low thermal conductivity impeded the transfer of heat to the underlying ice

(Cao et al., 2021). These combined effects slowed the ice melting process, thereby delaying the onset of the BUE. The wind speed had a relatively high influence on the FUS (11.0 %), primarily by promoting water mixing and increasing heat exchange between the atmosphere and the water, thus affecting the formation of lake ice. For the BUE date, while the effect of wind speed was primarily associated with mechanical cracking of the ice cover, especially when it was thin, Lake Ulansu's relatively low wind speeds tempered this influence, resulting in a modest influence of only 3.1 % on the BUE date

(Fig. 7).





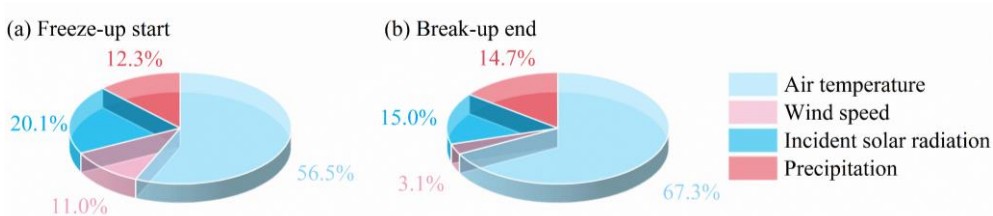

**Figure 7: Contributions of meteorological factors to Lake Ulansu's freeze-up start and break-up end dates.**

### 4.3 Monthly influence of meteorological factors on ice phenology

Although a holistic view of the contributions of meteorological factors to ice phenology is available in Fig. 7, it is crucial to understand that their influence is not equally important throughout the year. For example, air temperature, which is considered paramount (Bartosiewicz et al., 2021; Newton and Mullan, 2021), especially influences ice cover formation during its initial weeks to months. Therefore, detailed analyses revealing the impact of monthly variations in meteorological factors on ice phenology are necessary, and the results are shown in Fig. 8.

Our analysis, depicted in Fig. 8a, shows that November's air temperature correlated most strongly with the FUS dates ($R = 0.53$). This strong correlation can be attributed to the lake's shallow 1.6-m deep water, which allows rapid responsiveness to cold air. As air temperatures decrease, the gradient between the lake and the atmosphere increases, leading to continuous heat release from the water to the atmosphere (Lazhu et al., 2021). This dynamic causes the surface water temperature to decrease swiftly, increasing the density of the upper water layer and triggering vertical convection. The resulting sinking of the upper water layer expedites the cooling of the entire water body. This effect of air temperature is not confined to the FUS date; it extends to the BUE date (Fig. 8b), with the strongest correlation coefficients found for February ($R = -0.49$) and March ($R = -0.70$). The cumulative thickness of the ice cover during its formation period impacts the timing of the melting period. Therefore, the air temperature from November to March had some impact on the BUE date, with a $R > 0.3$.

Surprisingly, the correlation coefficient for solar radiation was notably low (Fig. 8a), and in November, it exhibited a negative correlation with the FUS date ($R = -0.37$). This trend is similar to the findings by Caldwell et al. (2020). For Lake Ulansu, the correlation might arise from the dependence of solar radiation absorption by water on the attenuation coefficient of light within water (Xie et al., 2023). Lake Ulansu is eutrophic (Yang et al., 2020) and contains numerous suspended particles and algae that absorb and scatter solar radiation. This high attenuation coefficient diminishes the penetration of solar radiation, leading to a muted heating effect on the water. Moreover, the incident solar radiation exhibited a decreasing trend, averaging between 119 and 125 W m$^{-2}$ in November. This not only limited the heating effect on the water but also suggests that the narrow range of variation might not capture the intricacies of its actual impact. Nonetheless, in January,





solar radiation became more influential ($R = -0.33$). Solar radiation heated the water body, increasing the ice–water heat flux

and promoting the melting of ice cover.

Wind plays a dual role in this process. While it can accelerate water mixing, promoting an early FUS, it can also disrupt

fragile ice, delaying the formation of a stable ice cover (Fig. 8a). At Lake Ulansu, where wind speeds generally range from 1

to 4 m s$^{-1}$, its influence on the FUS date was somewhat muted, as indicated by a $R$ of $-0.07$ in November. Similarly, the

effect of wind on the ice melting phase is marginal (Fig. 8b). Primarily impacting the ice surface, subdued wind speeds at the

lake contributed to a decrease in turbulent heat exchange and a consequent reduction in ice sublimation.

The role of precipitation is complex. When it manifested as rain over the lake in October, it cooled the water, narrowing the

temperature gap between the lake and the atmosphere ($R = -0.13$). Conversely, when it fell as snow in November, its low

thermal conductivity obstructed heat exchange between the lake and the atmosphere, hampering the formation of a stable ice

cover ($R = 0.27$). If there is a considerable amount of snow cover, the influence of air temperature and solar radiation on the

ice cover and the underlying water can be mitigated, but Lake Ulansu experienced limited snowfall, resulting in a minimal

impact on the BUE date (Fig. 8b).

Since ICD is the difference between the BUE date and the FUS date, the meteorological factors highly correlated with the

ICD were essentially the same for both the FUS and BUE dates (Fig. 8c). Air temperature remains the primary influencing

factor, with the correlation coefficient of solar radiation in January also being relatively high.




**Figure 8: Correlation coefficients between meteorological factors and ice phenology in different months. Significance levels for correlation coefficients are denoted by * ($p < 0.05$) and ** ($p < 0.01$).**

## 5 Discussion

### 5.1 Uncertainty analysis

Due to the specific characteristics of Lake Ulansu, the uncertainties in ice phenology reconstruction are mainly attributed to the spatiotemporal resolution constraints of the CETB data and the methodological limitations associated with the DMTT algorithm used in the study.

First, the CETB $T_b$ data had a high spatial resolution of 3.125 km. Due to the complex geometry of the shoreline and rich
aquatic vegetation of reeds in Lake Ulansu, five CETB grids with water coverage exceeding 0.70 were selected (Fig. 1b). The criteria for selecting these pixels were twofold: first, to ensure the inclusion of extensive water surfaces, thereby minimizing the influence of land surface $T_b$, and second, to locate the pixels predominantly in the central area of the lake where the water depth is more consistent, thus reducing the effects of uneven ice formation due to depth variation. Considering the overall shallow water of the lake, we contend that the potential error caused by our grid selection is minimal
when assessing the ice phenology of Lake Ulansu. Furthermore, despite a 1-day temporal resolution in the CETB data, periodic gaps in $T_b$ data may occur due to orbital issues. This problem is particularly pronounced in the SMMR sensor and could lead to a maximum error of approximately 5 days in ice phenology results for the years 1979–1987 (Cai et al., 2022). However, $T_b$ data from SSM/I and SSMIS are more complete, resulting in more accurate ice phenology data from these sensors (Fig. 5).

Second, the DMTT algorithm applied a smoothing technique to calculate the daily values of $T_b$ as the average of the preceding 10 days and the following 10 days, following Du et al. (2017). The purpose of this step was to eliminate the influence of short-term fluctuations on $T_b$ data but at the cost of losing some $T_b$ information. Furthermore, the DMTT algorithm used CETB data at 37.5 GHz. However, Kang et al. (2012) found that 18.7 GHz is more suitable for calculating the BUE date when analyzing phenological data for the Great Bear Lake and Great Slave Lake due to large changes in $T_b$
during the transition of ice into water, facilitating the detection of ACPs. This discrepancy may explain the slightly inferior BUE results in this study (Fig. 5). However, the coarse spatial resolution of 18.7 GHz in the CETB data is unsuitable for determining the shape of the shoreline of Lake Ulansu. In light of this, our study employed two thresholds to separately calculate FUS and BUE dates, obtaining thresholds better suited for the transition between ice and water states and thereby enhancing the accuracy of ice phenology results.



## 5.2 Comparison with other lakes

Lake Ulansu is characterized by its shallow depth and abundant aquatic vegetation, which is much different from most previous results on large lakes at similar latitudes and boreal lakes with heavy snowfall. Therefore, a comparison between these lakes and with the general trend across lakes in the Northern Hemisphere is interesting.

First, previous studies on lake ice phenology using passive microwave data were primarily limited by the coarse spatial resolution of $T_b$, thus focusing mainly on larger lakes. For example, Cai et al. (2017) used $T_b$ data to calculate ice phenology for Qinghai Lake from 1979 to 2016. They found that the FUS date delay and BUE date advancement were 0.16 d yr$^{-1}$ and 0.37 d yr$^{-1}$, respectively, which are weaker and stronger than those observed for Lake Ulansu. Importantly, Qinghai Lake is a massive saline lake, covering an area of more than 4,000 km$^2$, with a much larger water volume than Lake Ulansu. Therefore, during the freezing process, more heat must be released to bring the surface water temperature below 0 °C. This makes it relatively less sensitive than shallow lakes to air temperature fluctuations, hence resulting in a weak trend for the FUS dates. At altitudes greater than 3,000 m, Qinghai Lake's ice melting is markedly influenced by its exposure to intense solar radiation, leading to a more pronounced BUE trend (Kirillin et al., 2021). Ruan et al. (2020) employed a RF model and CMIP5 data to forecast ice phenology changes in lakes across the Tibetan Plateau from 2015 to 2099, yielding analogous outcomes. In the scenarios with the highest greenhouse gas emissions, the BUE trend was significantly greater than that of the FUS. In addition to its correlation with solar radiation, sublimation on the ice cover surface under strong winds also exerts a substantial impact on the BUE date. Huang et al. (2022) computed the mass balance of lake ice in the region and observed that sublimation accounted for 40 % of the maximum ice thickness loss. Various factors contribute to the pronounced BUE trends in lakes on the Tibetan Plateau. In contrast, long-term studies of Lake Ulansu indicate a stronger trend for the FUS than for the BUE (Table 2), with air temperature exerting an influence of over 50 % on both the FUS and BUE dates (Fig. 7).

Second, in the Northern European context, the impact of precipitation, particularly snow, is a significant modifier of ice phenology, as demonstrated in extensive studies of Estonian lakes by Nõges and Nõges (2013). Over the period from 1961 to 2004, despite the trend of rising air and lake surface temperatures, the BUE date displayed only minimal changes. This anomaly was attributed to the insulating properties of snow, which moderated the lakes' thermal response to climatic warming. The insulating effect of heavy snowfall was substantial enough to offset the rising air temperature's potential to advance the BUE date. In contrast, Lake Ulansu's region experiences markedly less winter precipitation (Table 3), resulting in minimal snow cover. The sparse snowfall, coupled with wind activity that disrupts snow accumulation, diminishes any insulative buffer on the ice surface, allowing air temperature to exert a more direct influence on the BUE date than Estonian lakes. Furthermore, research by Marszelewski and Skowron (2010) on six Polish lakes further supported the role of precipitation in ice phenology. They observed a rising trend in air temperatures in this region, resulting in an advancing trend for BUE dates. We believe that this is still closely related to precipitation. These lakes have relatively low water body heat capacities, similar to Lake Ulansu. Precipitation falling in the form of rain or snow on the lake surface accelerates the





cooling of water temperature and reduces the lake–atmosphere temperature difference. When a relatively thin ice layer forms on the lake surface, thick snow accumulates and presses columnar ice beneath the water surface. This generates negative

freeboard, driving the transformation of slush ice into snow ice (Cheng et al., 2020). This process still accelerates the formation of a stable ice cover. However, in the Lake Ulansu region, precipitation is relatively scarce, limiting the impact of precipitation on ice phenology.

Finally, ice phenology from 1931 to 2005 for 678 water bodies in the Northern Hemisphere, including North America, Europe, and Russia, was analyzed by Newton and Mullan (2021). They found that the ICD increased by 0.19 d yr$^{-1}$ between

1946 and 1975 and decreased by 0.58 d yr$^{-1}$ from 1976 to 2005. The average decrease in the ICD over the entire period was 0.06 d yr$^{-1}$. These trends in the ICD are consistent with those of Lake Ulansu during the same period (Table 2). It is essential to note that during the period from 1931 to 1975, more than 50 % of the lake regions experienced warming air temperatures (Newton and Mullan, 2021), aligning with the trend of increasing air temperature in Lake Ulansu from 1941 to 1982. Nevertheless, compared to the aforementioned lakes, Lake Ulansu displays more prominent trends in ice phenology. This is

mainly due to Lake Ulansu's distinctive lake features and regional climate conditions, which make it more responsive to air temperature changes.

## 6 Conclusion

To our knowledge, we reconstructed the ice phenology of Lake Ulansu from 1941 to 2023, the longest ice phenology data for a large, shallow, and aquatic plant-dominated lake in Northwest China. A new double-threshold moving $t$ test (DMTT)

algorithm was developed to distinguish water and ice from lake surfaces using 37 GHz H-polarization $T_b$ data with a resolution of 3.125 km from the CETB dataset. The DMTT has an integrated daily air temperature, which allows it to accurately detect abrupt change points (ACPs) and thus calculate distinct thresholds for transitions between water and ice. This enhanced functionality allowed us to effectively differentiate mixed pixels among complex shorelines and rich aquatic vegetation, resulting in an accurate ice phenology from 1979 to 2023. Together with the corresponding meteorological data,

these data further served as training data for the RF model, and the ice phenology of Lake Ulansu from 1941 to 1978 was successfully reconstructed.

Over the 83 years from 1941 to 2023, Lake Ulansu exhibited an average FUS date of 15 ± 5 November and an average BUE date of 25 ± 6 March, with an average ICD of 130 ± 8 days. The trends in FUS, BUE dates, and ICD were 0.07 d yr$^{-1}$, −0.01 d yr$^{-1}$, and −0.08 d yr$^{-1}$, respectively. Notably, we did not find significant trend changes in the delayed FUS and advanced

BUE dates, which differs from the ice phenology results for other lakes. This phenomenon can primarily be attributed to significant air temperature fluctuations in the 1980s, characterized by a cooling trend before the 1980s and a warming trend thereafter. In stark contrast to these slight temporal shifts, the last four decades have witnessed a considerable contraction in the ice cover span, with the ICD shortening by an average of 22 days. The predominant role of air temperature in ice phenology has been confirmed, showing it contributes 56.5 % of the variation in the FUS date and 67.3 % in the variation in

the BUE date. Seasonal transitions further underscore the role of air temperature, particularly in November, where it strongly correlates with the FUS date ($R = 0.53$) and extends into early spring, influencing the BUE date ($R = -0.50$). The high eutrophication level in Lake Ulansu attenuates the capacity of solar radiation to heat the waters, with the most substantial influence observed during the ice melting process in January, as evidenced by a correlation coefficient of $R = -0.36$. Wind speed, while contributing to water mixing and early ice formation, has a limited effect on the FUS date and an even lesser

impact on the BUE date due to the lake's generally low wind conditions. The role of precipitation is complex, with rain cooling the water and snow impeding heat exchange because of its low thermal conductivity, yet both have a minimal overall impact on the BUE date in light of the lake's limited snowfall. Overall, wind speed, precipitation, and solar radiation collectively accounted for 43.5 % of the influence on the FUS date and 32.7 % on the BUE date, highlighting the multifaceted yet secondary roles of these factors in shaping the ice phenology of Lake Ulansu compared to the predominant

role of air temperature.

This study proposed feasible methods to significantly enhance the capability for ice phenology studies in lakes with complex shorelines and abundant vegetation, offering reliable technical support for ice phenology analysis in similar lakes across the Northern Hemisphere. Additionally, the data gaps in early satellite records, particularly those of Nimbus 7, warrant further investigation to achieve a more comprehensive historical perspective. Addressing these gaps is crucial for enhancing the

accuracy of long-term ice phenology reconstructions. We also expect future research to delve into the composition, chemical properties, and microstructure of lake ice through numerical simulations, thereby opening new possibilities for remote sensing applications across various lake types.

**Appendix A: Adaptability of the DMTT algorithm in the Tb series**

The DMTT algorithm shows strong adaptability in identifying the $T_b$ series from mixed pixels. Although these series

maintain a 'W' shape overall, there are differences in the specific $T_b$ during the freezing and melting stages. Figure A1a show smaller changes in brightness temperature between these two stages, and Fig. A1b show a lower $T_b$ valley during melting than during freezing. Such sensitivity to nuanced changes underscores the substantial advantage of the DMTT algorithm in enhancing the precision and continuity of long-term ice phenology data, especially in environments with complex shorelines and extensive aquatic vegetation.




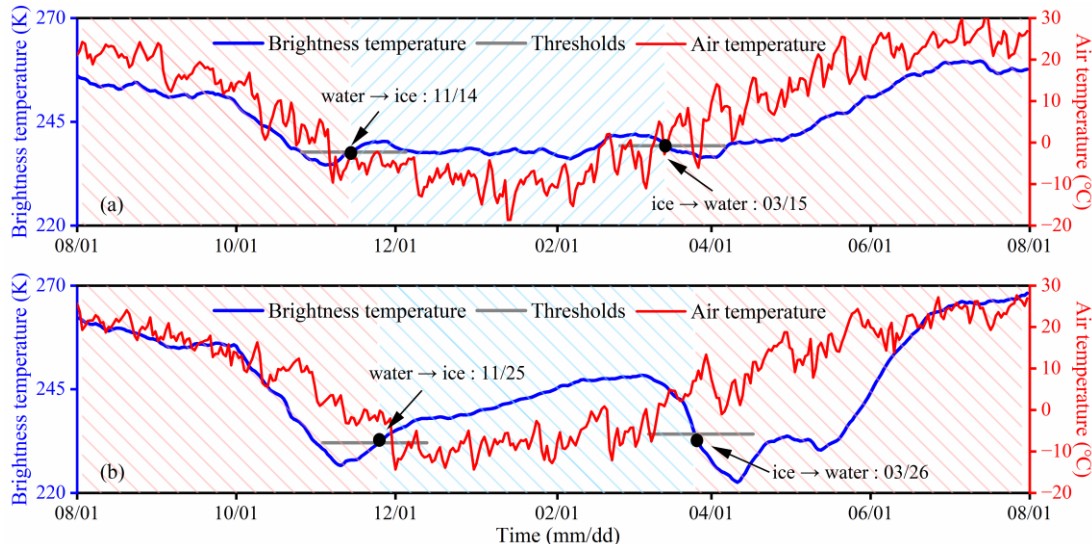

**Figure A1: Brightness temperature (blue solid line) and air temperature (red solid line). The ice/water status was determined using the double-threshold moving *t* test algorithm, where the red and blue shaded regions represent the water and ice states, respectively. (a) for 2001 and (b) for 2015.**


## Appendix B: Comparative validation with Du et al. (2017) and field observations

Du et al. (2017) utilized the MTT algorithm to calculate the ice phenology of Lake Ulansu over a 14-year period from 2002 to 2015 (https://nsidc.org/data/nsidc-0726/versions/1#anchor-2). However, it was only possible to effectively extract data for 5 to 8 years (Fig. B1). This limitation primarily arose from the challenges of the MTT algorithm in handling the complex surface conditions of lakes, especially when the surface was composed of mixed pixels containing aquatic vegetation or land.

These mixed pixels led to approximately 49% of the days in the MTT algorithm's time series being incorrectly classified as states without detected ice or water, significantly impacting the continuity and accuracy of long-term ice phenology data. In contrast, we developed the DMTT algorithm, which has demonstrated improved handling of these complexities, reducing undetected states and substantially enhancing the accuracy of the dataset. Despite the scarcity of field observations, these

data have proven to be invaluable, playing a pivotal role in validating the precision of the DMTT algorithm.



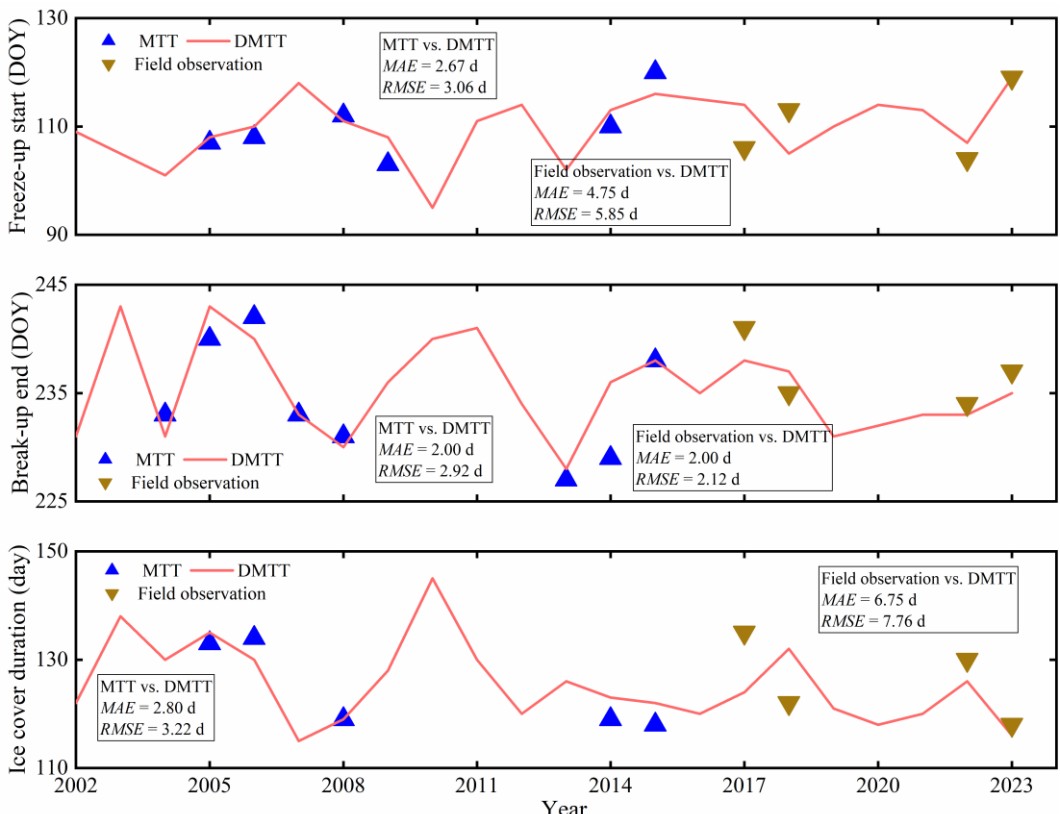

**Figure B1: Comparison of the ice phenology for Lake Ulansu as extracted by the DMTT algorithm (2002–2023) with that by the MMT algorithm (2002–2015) and field observations (2017, 2018, 2022, 2023).**


## Appendix C: Optimal number of trees for the random forest model of ice phenology

In the process of hyperparameter optimization for our random forest model, we carefully considered the balance between model complexity and predictive accuracy. A greater number of trees, such as 100, might yield a lower *MAE* on the training set, which is also associated with a greater risk of overfitting. Overfitting occurs when a model learns the training data too well, including noise and outliers, which results in decreased performance on unseen data, as evidenced by a larger *MAE* on

the validation set (Table C1).

By selecting 20 trees, we strike an optimal balance where the model complexity is sufficient to capture the underlying patterns in the data without being overly sensitive to the noise in the training data. We observed that increasing the number of trees from 20 to 100 did not significantly improve the *R* on the training set. Moreover, for the FUS model, there was a

notable decrease in *R* on the validation set when 50 or 100 trees were used compared to 20 trees. This indicates that the



additional complexity introduced by more trees may lead to overfitting, which harms the model's predictive performance on new data. Moreover, the performance on the validation set for models with 20 trees suggests robust generalization without significant overfitting, as indicated by consistent $R^2$ and acceptable *MAE* and *RMSE* values. Therefore, considering the potential for overfitting, the computational efficiency, and the model's generalization performance, we concluded that a

configuration of 20 trees is the most appropriate for our study.

**Table C1. Performance of freeze-up start date and break-up end date predictions on training and validation sets with varying numbers of trees in a random forest model.**

| Number of trees | Ice phenology | Training | | | Validation | | |
|---|---|---|---|---|---|---|---|
| | | *R* | *MAE* (d) | *RMSE* (d) | *R* | *MAE* (d) | *RMSE* (d) |
| 10 | Freeze-up start | 0.94 | 1.62 | 2.28 | 0.92 | 3.40 | 4.32 |
| | Break-up end | 0.93 | 1.56 | 2.21 | 0.81 | 4.76 | 5.37 |
| 20 | Freeze-up start | 0.96 | 1.35 | 1.87 | 0.94 | 3.21 | 3.85 |
| | Break-up end | 0.94 | 1.55 | 2.02 | 0.80 | 4.74 | 5.37 |
| 50 | Freeze-up start | 0.97 | 1.31 | 1.99 | 0.93 | 3.84 | 4.68 |
| | Break-up end | 0.96 | 1.43 | 1.80 | 0.78 | 4.93 | 5.60 |
| 100 | Freeze-up start | 0.98 | 1.23 | 1.82 | 0.95 | 3.63 | 4.33 |
| | Break-up end | 0.96 | 1.44 | 1.83 | 0.78 | 4.84 | 5.59 |

**Data availability**

The ice phenology data for Lake Ulansu can be accessed at https://doi.org/10.5281/zenodo.10848522. The CETB data were downloaded from the NSIDC Data Search (https://nsidc.org/data/nsidc-0630/versions/1). The ERA5 data were downloaded from the Climate Data Store (https://cds.climate.copernicus.eu/).

**Author Contributions**

PZH and PL conceived the study and developed the theoretical framework (Conceptualization). Funding was secured by PL, QKW, and XWL (Funding acquisition). The data analysis and visualization were performed by PZH, PL, and BC (Formal analysis and Visualization). PL and BC provided guidance and oversight throughout the research process (Supervision). The initial draft of the manuscript was written by PZH (Writing – original draft preparation). All authors contributed to subsequent revisions (Writing – review & editing).

P.H. and P.L. conceived the study and developed the theoretical framework (Conceptualization). Funding was secured by P.L., Q.W., and X.L. (Funding acquisition). The data analysis and visualization were performed by P.H., P.L., and B.C. P.L.



and B.C. provided guidance and oversight throughout the research process. The initial draft of the manuscript was written by P.H. All authors contributed to writing, editing, and reviewing.

## Competing interests

At least one of the (co-)authors is a member of the editorial board of *The Cryosphere*.

## Financial support

This work was funded by the National Key Research and Development Program of China (Grant No. 2023YFC2809102); the National Natural Science Foundation of China (Grant No. 42320104004); the National Key Research and Development Program of China (Grant No. 2022YFE0107000); the National Natural Science Foundation of China (Grant No. 42106233).

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
