# Peer review of "Reconstructing ice phenology of lake with complex surface cover: A case study of Lake Ulansu during 1941–2023"

_EGUsphere, 2024_

## Referee Comment (RC2)

**General comments**

Lake ice phenology is a sensitive indictor to climate change.

The paper proposed a new double-threshold moving t test (DMTT) algorithm applied to CETB dataset, and reconstructed the ice phenology of Lake Ulansu from 1941 to 2023 using Random Forest based on the ERA5 data, which have been proved relatively good performance. The method is novel in identifying lake ice phenology and the influencing factors, but there still need some improvement.

(1) The introduction is not well organized, and need to improve.

(2) The logic of methodology is not clear. The overview of Data makes the reader confusing.

(3) The caption of figures should be concise, and the author need to check and improve them.

(4) The correlation between solar radiation and lake ice phenology is lower than we expected. The authors analyzed this in terms of month. Freeze-up and break-up date are one value in a winter, and other parameters varies each month, how to connect them together?

In conclusion, the subject of the study is suitable for "The Cryosphere" and can potentially be of interest for the journal's wide audience, and we suggest a major revision.

**Specific comments**

The Introduction is not clear and attractive. The meaning of long-term tracking of climate records such as lake ice phenology, is not well expressed. The research meanings in the first paragraph of Introduction are not closely related with the content, and the author should rewrite. The scientific problem and study goals are note clear.

Line 38 the best quality is not appropriate.

Line 41 MODIS just mentioned 8-day product, and the author should update recent work.

Line 46 The active microwave is not suitable for the large lakes. This sentence should be improved, and supplement more contents.

P47 One advantage of passive microwave is frequent revisit, which is not mentioned.

The second paragraph is too long, and should be divided. The core of this work belongs to passive microwave, and could be discussed alone.

The fitting methods of lake ice phenology based on climate records (such as air temperature) are note well discussed, just listed some previous works.

The author explained how the study organized, and the goals of this study is not clear. The last paragraph should be rewritten.

The general description of ice regime should be added in Study are, like ice thickness, regular winter recreation.

Line 75 300 need to check, or provide the citation.

Line 80 how to exclude the influence of vegetation?

Line 83-84 remove the date.

Line 85 The definition of hydrological year is not accurate. For example, HY 2022 lasted from August in 2021 to July in 2022.

Line 90 The title of figure 1need to revise.

Line 95-105 The logic of flowchart is not clear, and Figure 2 should be revised. The input and output is not clear. The different colors have certain meanings? The method is too short.

Line 119 high spatial resolution Tb data are essential, gramma error

Line 124 These data were sourced from the SMMR on the Nimbus 7 satellite; the SSM/I on the F08, F10, F11, F13, F14 and F16 satellites; and the SSMIS on the F16 satellite. Move this to the above content.

Figure 3 have two choices: (1) make the content short and concise, and add the figure in the supplementary materials; (2) move this part to Results.

Figure 4 "between 1 August and 31 July from 1979 to 2023 for Lake Ulansu": check this.

"The solid lines represent the interannual average brightness temperature and air temperature time series, respectively. " delete this sentence. Lack the description of Figure 1 (a), just (b) appears.

"

Line 174 "proportion lake water greater than 0.70" how the thresholds are determined? Why do not use 0.8/0.2?

Line 236 "Seventy percent" and "30 %", keep the same expression.

Line 258 ERA5 have various types of climate data, which one you used? It is not clear. The input and output of RF is confusing.

Figure 6 Please add the math equation and basic index of linear regression. Please add the lengend of different lines, it would be better to remover the description of dashed lines. (a)-(f) explained separately.

Line 280 add the yearly changing rates.

Line 282 The title of 4.2 is not proper.

Table 3 Linear tread is yearly changing rates? Please add the unit. The linear tread of wind speed is 0. Check this?

Figure 8 The correlation between solar radiation and lake ice phenology is lower than we expected.

As for random forest, we have following questions:

(1) The abbreviation of correlation coefficients is r, not R. Need to check the whole manuscript.

(2) The evaluate the performance of RF, the determination coefficients $R^2$ is more usually used, rather than R.

(3) How the author avoid overfitting in the work? Need to explained more clearly.

(4) Why the author chose random forest rather than other methods?

The two paragraphs in Author contribution are repeatedly expressed.

---

## Author Comment (AC1)

Reply to RC1

Note: the comments and authors' replies are in black and blue font, respectively. All changes in the revised manuscript are highlighted in yellow.

This manuscript had two major goals. First to create an improved algorithm for extracting breakup and freeze-up dates using microwave-based temperature brightness data from satellites on a shallow, and vegetated lake. And second, to re-construct a long ice phenology timeseries extending back in time before the availability of satellite-based estimates. The novelty of the study appeared to be the development of the DMTT algorithm to extract freeze-up and break-up dates. The DMTT algorithm appears to be better suited for mixed pixels more commonly associated with lakes with vegetated cover or complex shorelines.

The authors could use more clarity in their introduction and methods sections, specifically to both highlight the novelty of the study, summarize what has been done in the past, and differentiate better the different types of data. Regarding novelty, I think the authors should spend more time comparing the DMTT and MTT (e.g. expanding more on the improvements made). I believe this will help readers understand which algorithm to use given their own lake data. Further, the authors state that their goal was to understand shallow lakes with complex shorelines and mixed pixels, however they chose 5 pixels from the CETB dataset which only contains pixels that are mostly water. Also, the pixels are not fine resolution enough to capture the complex shoreline. The authors could spend more time justifying the use of this type of data for ice phenology research, especially on shallow lakes with complex shorelines/vegetation, or de-emphasize the mixed pixels and focus more on the shallow lake aspect. In the same vein, I suggest the authors spend some time in their introduction qualifying why microwave-based methods are more accurate or better than optical based methods. One suggestion would be to discuss if microwave data can improve upon the errors associated with missing the freeze-up or break-up dates due to cloudy pixels, which is ~5 days for the MODIS method (Zhang and Pavelsky, 2019) or whether microwave data better corresponds to in-situ data, if available. Finally, the authors could more clearly differentiate what data were used and where they came from. I liked the idea of Figure 2, but I found it hard to follow. I would suggest starting with raw data on the left and moving to more derived products on the right (or the same suggestion but top to bottom). All external validation data should be clearly marked versus the data use for testing and training in the RF model. I put more detailed comments about clarity throughout the

methods section.

Reply: Thank you very much for your constructive comments on our manuscript. We have addressed each of your concerns and please see our response to the point of view listed above:

(1) We have thoroughly restructured the Introduction (lines 23-57) to better articulate the importance of studying ice phenology in shallow, vegetated lakes and to emphasize the advantages of passive microwave remote sensing over other methods.

(2) The data description (Section 3.1) and methodological details (Section 3.2) have been revised for clarity. Each dataset's purpose and application within our study are now explicitly defined, underscoring the rationale behind our choices and their relevance to the study objectives. We have also introduced a new section (4.1 Algorithm evaluation) that not only compares our results with optical remote sensing outcomes but also with MTT algorithm outputs and in-situ observations, demonstrating the robustness and superiority of the DMTT algorithm.

(3) We provide comprehensive justifications for selecting predominantly water pixels, highlighting their relevance in capturing valuable ice phenology insights even in mixed pixels (detailed response in specific comments 20-23). This selection is crucial given the complex shorelines and vegetated cover of Lake Ulansu, demonstrating why the DMTT algorithm is particularly suitable for our study conditions. Furthermore, in Appendix F, we have included examples of similar "$\Omega$"-type brightness temperature series from other lakes to illustrate the broader applicability and current research gaps in lake ice phenology under algorithm constraints.

(4) We have clarified the advantages of using passive microwave remote sensing over optical methods in our study. The passive microwave data, which cover a more extended period and offer daily revisit capabilities, were instrumental in reconstructing the ice phenology from 1979 to 2023. This long-term dataset was compared against the shorter-term optical remote sensing data from MODIS, Landsat and Sentinel-2 from 2000 to validate our findings, as detailed in the revised Section 3.1.2 Optical satellite data.

(5) Figure 2 has been redesigned for improved clarity. It now presents a logical progression from raw data to processed outputs in a top-to-bottom format, making it easier for readers to understand the data workflow used in our study. This figure effectively delineates the types of data used for training and testing in the random forest model versus those used for external validation.

[Figure]

**Figure 2: Reconstruction of the ice phenology of Lake Ulansu (1941–2023) based on ERA5, CETB, and optical satellite data via the double-threshold moving *t* test algorithm and random forest model.**

Specific comments:

1. Line 10 is this multi-source? Can you be more specific? Is this ERA5 data or other types of data?

Reply: We utilized Calibrated Enhanced-Resolution Passive Microwave Daily EASE-Grid 2.0 Brightness Temperature (CETB) data. These data were sourced from the SMMR on the Nimbus 7 satellite; the SSM/I on the DMSP-F08, -F10, -F11, -F13, and -F14 satellites; and the SSMIS on the DMSP-F16 satellite. Meteorological data were obtained from ERA5. To clarify the abstract, it has been revised to "*To address this challenge, a new double-threshold moving t test (DMTT) algorithm, which uses SMMR and SSM/I-SSMIS sensor-derived brightness temperature data at a 3.125-km resolution and long-term ERA5 data, was applied to capture the ice phenology of Lake Ulansu from 1979 to 2023.*"

2. Lane 23 consider "global lake ice loss is a prominent feature associated with climate change" or simplifying this beginning sentence so it is in more active voice.

Reply: Revised accordingly.

3. Line 25 what do you mean by ecological balance?

Reply: We have revised the sentence: "*Lake ice regulates water temperature, light availability, and nutrient circulation, which are crucial for the ecological environment (Latifovic and Pouliot, 2007; Wu et al., 2022).*"

4. Line 26 can you elaborate on how ice cover impacts the momentum and mass exchange between the atmosphere and water?

Reply: Ice cover in lakes plays a crucial role in regulating the exchange of momentum and mass between

the atmosphere and the water body. The following is an elaboration of this process:

(1) Momentum exchange: The presence of ice on the lake surface acts as a barrier, altering momentum exchange by reducing the direct transfer of wind energy to the water. This results in lower surface stress and reduced mixing within the water column. The momentum from the wind is partially transferred to the ice, causing it to drift, but the movement and mixing of the water below the ice are minimal compared with those in ice-free conditions. In the absence of ice, the wind directly impacts the water surface, creating waves and turbulence, which enhances the mixing of the water column. This mixing is critical for the distribution of heat, nutrients, and other materials throughout the water body (Kirillin et al., 2012; Leppäranta et al., 2023).

(2) Mass exchange: Ice cover reduces evaporation rates, limiting the transfer of water vapor from the lake to the atmosphere. This affects local humidity levels and can influence weather patterns. Additionally, ice can trap gases such as methane and carbon dioxide beneath it, altering gas exchange dynamics and impacting the lake's biogeochemical cycles (Zhao et al., 2022; Li and Xue, 2021).

We appreciate this opportunity to elaborate on these processes, enhancing our understanding of the broader environmental impacts of lake ice cover. Given the extensive revisions made to the Introduction of our manuscript to enhance clarity and focus, this detailed discussion of ice cover impacts on momentum and mass exchange was not included in the main text to maintain coherence and relevance to the primary research aims.

5. Line 28 see also Sharma et al. 2019, consider wording similar to: Many studies have evaluated lake ice phenology trends in the northern hemisphere, but shallow and vegetated lakes remain largely unexplored due to their lack of observational data

Sharma, Sapna, Kevin Blagrave, John J. Magnuson, Catherine M. O'Reilly, Samantha Oliver, Ryan D. Batt, Madeline R. Magee, et al. 2019. "Widespread Loss of Lake Ice around the Northern Hemisphere in a Warming World." Nature Climate Change 9 (3): 227–31. https://doi.org/10.1038/s41558-018-0393-5.

Reply: We rephrased this sentence to "*Many studies have evaluated lake ice phenology trends in the Northern Hemisphere (Mishra et al., 2011; Sharma et al., 2019; Woolway et al., 2020), but shallow and vegetated lakes remain largely unexplored owing to their lack of long-term observational data.*"

6. Line 32 The sentence "shallow lakes are more sensitive to climate change then deep lakes because of ...., because of their potential ecological effects on algal blooms" does not seem like a complete sentence. Are you trying to say that the fact that these lakes get algal blooms mean that they are more

sensitive to climate change. Lake ice and ice formation seem to clearly stem from climate change sensitivity, but I feel that algal bloom formation is related to several factors including temperature, ice, nutrients, etc.

Reply: Here, the revised version addresses your concerns: "*These lakes are more sensitive to climate change than deep lakes are because of their low heat capacity (Ambrosetti and Barbanti, 1999), which makes them more responsive to temperature fluctuations. A shortened ice-covered season leads to a longer period of open water, enhancing conditions for algal bloom growth and potentially causing ecological imbalances (Duan et al., 2012).*"

7. Line 34 what do you mean by microlevel, microscopic, lake-specific, etc.?

Reply: In the context of our manuscript, "microlevel" interactions refer to small-scale, detailed interactions within lake ecosystems that encompass physical and biological processes.

To make the text clearer, we revised it to "*Ice phenology data from such shallow lakes are crucial for understanding how temperature fluctuations influence water stratification, nutrient availability, and the biological rhythms of aquatic organisms (Sharma et al., 2021; Smits et al., 2021).*"

8. Line 37 Consider Hampton et al. 2017 (or citations within) if you want to make a broad statement of the need more lake ice research irrespective of shallow lakes, otherwise make this statement more explicitly about shallow lakes

Hampton, Stephanie E., Aaron W. E. Galloway, Stephen M. Powers, Ted Ozersky, Kara H. Woo, Ryan D. Batt, Stephanie G. Labou, et al. 2017. "Ecology under Lake Ice." Edited by James Grover. Ecology Letters 20 (1): 98–111. https://doi.org/10.1111/ele.12699.

Reply: Revised accordingly.

9. Line 47 what is the temporal resolution?

Reply: We added this information to the Introduction: "*Active microwave remote sensing is utilized primarily for extracting ice phenology data from large water bodies because of its relatively low (> 10 days) temporal resolution (Antonova et al., 2016; Howell et al., 2009).*"

10. Line 57 consider rewording the sentence that starts with "in addition…" This sentence marks an important transition from your text about creating phenology records using satellite data and creating a forecasting/back casting model using meteorological data. Here maybe you could emphasize past work

using satellite (optical/microwave) as training data to create longer term time series

Reply: To make the text clearer, change it to "*In addition, if long-term lake ice phenology records are desired, satellite remote sensing can also provide training datasets for machine learning (Wu et al., 2021; Xu et al., 2024). For example, Ruan et al. (2020) presented a practical machine learning application in which a random forest model was used to forecast the ice phenology of lakes on the Tibetan Plateau until 2099. The ice phenology history derived from passive microwave remote sensing and CMIP6 meteorological data was employed here as inputs. The success of this application largely depends on the quality of the training data, which requires not only accuracy but also a long temporal record that captures the variability of environmental conditions. To ensure the robustness of the model, it is critical to integrate diverse meteorological factors, such as air temperature, wind speed, and precipitation, along with ice phenology data, as these factors significantly influence lake ice. Furthermore, developing effective machine learning models that can address the complexities of lakes with various surface characteristics is essential.*"

11. Line 68 using which data?

Reply: The original text has been modified to "*A new algorithm was developed to classify the ice and water states on the lake surface in brightness temperature data from the SMMR and SSM/I-SSMIS sensors for the period 1979–2023.*"

12. Line 70 using the created data in step one as training data?

Reply: Yes. To make the text clearer, change it to "*(2) A random forest model was trained using the results in step (1) to reconstruct the ice phenology from 1941 to 1978.*"

13. Line 80 Is there a citation associated with the evaporation and precipitation data?

Reply: Two corresponding citations have been added. See details on lines 84-86.

14. Line 100 Am I correct that the phenology record using CETB was not used in the training of the random forest model, only the phenology record derived from optical satellite data?

Reply: Ice phenology data from 1979 to 2023, derived from CETB data, were used to train the random forest model. The ice phenology data obtained from optical satellite data (2000 to 2023) were used to validate the results derived from the brightness temperature data (1979 to 2023) but were not used for training the random forest model. We have made the necessary changes in the manuscript to reflect this

clarification. See details on line 106.

15. Line 100 meteorological data from ERA5?

Reply: Yes, meteorological data from ERA5 for the period of 1979 to 2023 were used to train the random forest model. Additionally, ERA5 data for the period from 1941 to 1978 were used to calculate historical ice phenology data. We have clarified this in the manuscript to avoid any confusion. See details on lines 107-109.

16. Line 113 What is the spatial resolution

Reply: We added spatial resolution information in Section 3.3.1, "*The CETB data were obtained from various satellite sensors with coarse spatial resolutions (~25 km), including SMMR on Nimbus 7, SSM/I and SSMIS on the DMSP satellite series, and AMSR-E on Aqua (Brodzik et al., 2016).*"

17. Line 125 the ice phenology data originated from the cloud bitmask in the state_1km product of MODIS? How does that reduce the influence of clouds?

Reply: Huo et al. (2022) calculated the ice phenology data for Lake Ulansu after 2000 via optical satellite data, specifically the red band reflectances from the MOD09GQ and MYD09GQ datasets, cloud information from the state_1 km_1 parameter in the MOD09GA and MYD09GA datasets, and the Landsat and Sentinel-2 datasets. The temporal resolution of both MOD and MYD is one day, with images taken at 10:30 AM and 1:30 PM local time, respectively. To fill real pixels under clouds, the higher temporal resolution of the MODIS products was utilized by merging MOD09GQ from the Terra satellite with MYD09GQ from the Aqua satellite. The details are as follows:

The first step to fill the cloud-covered pixels is shown in Equation (R1), where $P$ denotes the pixels and the output is the pixel type (water/ice) of the output satellite. It is divided into two cases: (1) if Aqua data are identified as water (ice) at pixel $A$, a pixel is defined as water (ice); (2) if Aqua data are identified as clouds at pixel $A$ and the Terra satellite is identified as water (ice) at pixel $A$, a pixel is defined as water (ice).

$$P_{(A,\text{output})} = \begin{cases} \text{water/ice}, & \text{if } P_{(A,\text{Aqua})} = \text{water/ice} \\ \text{water/ice}, & \text{if } P_{(A,\text{Aqua})} = \text{cloud AND } P_{(A,\text{Terra})} = \text{water/ice} \end{cases} \quad (R1)$$

The second step uses the temporal continuity principle to associate the type of $A$ pixel on day $t$ with the same pixel on days $t-1$ and $t+1$ (Equation R2). The detail is that if the $A$ pixel on day $t$ is a cloud pixel

and is discriminated as water (ice) on days $t-1$ and $t+1$, then the pixel on day t is defined as water (ice). The formula is as follows:

$$P_{(A,t)} = \text{water/ice}, \qquad \text{if } P_{(A,t-1)} = P_{(A,t+1)} = \text{water/ice} \tag{R2}$$

Some cloud pixels that exist after Equation (R2) is applied are further calculated on the basis of the five days of temporal continuity. As shown in Equation (R3), there are two cases: (1) if the $A$ pixel is a cloud on days $t-1$ and $t$ and is water (ice) on days $t-2$ and $t+1$, then the $A$ pixel is defined as water (ice) on days $t-1$ and $t$; (2) if the $A$ pixel is a cloud on days $t$ and $t+1$ and is water (ice) on days $t-1$ and $t+2$, then the $A$ pixel is defined as water (ice) on days $t$ and $t+1$.

$$\begin{cases} P_{(A,t)} = P_{(A,t-1)} = \text{water/ice}, & \text{if } P_{(A,t-2)} = P_{(A,t+1)} = \text{water/ice} \\ P_{(A,t)} = P_{(A,t+1)} = \text{water/ice}, & \text{if } P_{(A,t+2)} = P_{(A,t-1)} = \text{water/ice} \end{cases} \tag{R3}$$

Based on the above three steps, Huo et al. (2022) reduced the average cloud coverage from over 30% to 6.4%. This study uses the ice phenology data for Lake Ulansu calculated by Huo et al. (2022) to validate the passive microwave remote sensing results. Therefore, the detailed cloud removal process is not extensively explained in the manuscript. To ensure logical consistency in Section 3.1.2, "*These optical satellite data include red band reflectances from the MOD09GQ and MYD09GQ datasets, cloud information from the state_1 km_1 parameter of the MOD09GA and MYD09GA datasets, and the Landsat and Sentinel-2 datasets. The single-band threshold method classifies red band reflectances into water and ice pixels. Compared with higher spatial resolution Landsat and Sentinel-2 datasets, the dynamic threshold method aims to determine the optimal threshold for distinguishing ice and water pixels (Zhang and Pavelsky, 2019). Additionally, the spatiotemporal continuity of MODIS datasets is used to fill in real pixels under clouds, resulting in more accurate pixel classification.*

18. Line 131 Which part of Sentinel-2 and Landsat did you use to create your threshold? Can you elaborate more on the specifics of this method (e.g. Fmask on Landsat, and a threshold of X in the MODIS red band reflectance from the 250 m product)

Reply: The red band reflectances from MOD09GQ and MYD09GQ were used to differentiate between ice and water pixels (Zhang and Pavelsky, 2019), specifically via Equation R4:

$$P = \begin{cases} \text{ice}, & \text{if Band } 1 \geq a \\ \text{water}, & \text{if Band } 1 < a \end{cases} \tag{R4}$$

where Band 1 represents the reflectance of the red band and *a* represents the threshold to distinguish between water and ice.

Then, the Normalized Difference Snow Index (NDSI) formula was subsequently applied to Landsat and Sentinel-2 images to distinguish between ice and water pixels:

$$NDSI = \frac{\text{Band 2} - \text{Band 5}}{\text{Band 2} + \text{Band 5}} \quad (R5)$$

where Band 2 and Band 5 represent the reflectances of the green and shortwave infrared bands, respectively, in the Landsat and Sentinel-2 products. NDSI = 0.4 was selected as the threshold (Cai et al., 2019). The vector boundary of water/ice was extracted via manual visual judgment, and the water/ice coverage in the Landsat/Sentinel-2 images was calculated.

Finally, the dynamic threshold method was employed to constantly modify the threshold *a* to change the water/ice classification results of the MOD09GQ/MYD09GQ dataset and compared with the water/ice coverage areas of the Landsat and Sentinel-2 products with high spatial resolutions to find the optimal threshold *a*.

Given that this section is not the core focus of our study, these formulas are not presented in Section 3.1.2.

19. Line 135 Did you use the phenology record directly from Huo et al. 2022? If so, then I would put that citation further up

Reply: This study directly uses the results of Huo et al. (2022) to validate the ice phenology results derived from brightness temperature ($T_b$) data. To make the purpose of Section 3.1.2 clearer, it is now stated in the first sentence of the section "*In this study, we use ice phenology data for Lake Ulansu from 2000 to 2023 obtained from optical satellites by Huo et al. (2022) to validate the ice phenology results derived from $T_b$ data.*"

20. Line 179: Did you mean "we could not use the omega-shaped analysis for this study because the "w" shape violates that method"? The sentence as it currently reads is a little vague as to why the omega series are not available

Reply: The primary issue is the W-shaped variation in the brightness temperature ($T_b$) series at Lake Ulansu, which is significantly influenced by mixed pixels, unlike the pure water pixels observed in larger lakes such as Great Bear Lake and Qinghai Lake, which display a clear $\Omega$-shaped $T_b$ series (Fig. R1).

[Figure]

**Figure R1: Example of an Ω-shaped $T_b$ series in a large lake. (a): Great Bear Lake (Cai et al., 2022). (b): Qinghai Lake (Su et al., 2021).**

The specific challenges presented by Lake Ulansu's $T_b$ series are as follows:

(1) Land influence: Land areas, with their lower heat capacity than water does, respond more quickly to air temperature fluctuations. This rapid response causes significant changes in surface temperature, leading to greater fluctuations in the $T_b$ series (Munn and Richards, 1963; Wilheit et al., 1972).

(2) Reduced $T_b$ changes: At the critical moments of freezing and melting, the $T_b$ changes in mixed pixels are less pronounced (< 20K) than those in pure pixels (> 50K, Fig. R1), necessitating more precise threshold settings for correct classification.

(3) Increased abrupt change points: The variability in $T_b$ due to the land response to air temperature increases the number of abrupt change points (ACPs) detected by the moving $t$ test (MTT) algorithm, leading to potential misclassification of ice and water.

(4) Necessity for dual thresholds: Pure water pixels exhibit stable changes in $T_b$, allowing the use of a single threshold to calculate freeze-up start and break-up end dates. In contrast, the varying surface temperature and emissivity in mixed pixels require separate thresholds for accurate classification.

Owing to these complexities, the Ω-shaped analysis method is not applicable. To address this, we

developed a double-threshold moving $t$ test (DMTT) algorithm to extract ice phenology from the W-shaped $T_b$ series, accommodating the mixed pixel conditions of Lake Ulansu.

Revised manuscript text "*In this study, the ice phenology from 1979 to 2023 was determined by the brightness temperature ($T_b$) of the lake surface because $T_b$ changes considerably when a phase change occurs on the lake water surface (Su et al., 2021), providing a clear reference for determining the onset and end of the ice period. However, owing to the complex shape of the Lake Ulansu shoreline, which is characterized by its narrow form, pixels encompass not only water but also aquatic vegetation and land. As a result, five $T_b$ grids with a proportion of lake water greater than 0.70 were selected to represent the status of the lake surface (Fig. 1b). Figure 3a shows the annual variation in $T_b$ for each grid of each year as blue dashed lines, with the blue solid line showing the average for all grids during 1979–2023 and the solid red line depicting the average air temperature. The annual variation in the $T_b$ of Lake Ulansu exhibited a typical 'W' shape (Fig. 3a). This is different from what is observed for large lakes with only water surfaces where $T_b$ is typically a pure pixel, resulting in a line with an 'Ω' shape (Cai et al., 2022; Su et al., 2021). Compared with those of pure pixels, the mixed pixels of Lake Ulansu lead to inconsistent changes in surface temperature and emissivity, complicating the $T_b$ series. To accurately determine the ice phenology, it is necessary to precisely establish different thresholds for the transition between water and ice. Thus, previous methods to address the Ω-shaped $T_b$ series are not available in this study, and a double-threshold moving t test (DMTT) algorithm was developed to extract the ice phenology from the W-shaped $T_b$ series.*"

21. Line 180: Can you use this algorithm from all W-shaped series? How common is W-shaped series?

Reply: In this study, we focused exclusively on Lake Ulansu, which exhibited mixed pixels and a W-shaped $T_b$ series due to its complex shoreline and seasonal freeze–thaw patterns. This W-shape is characteristic of lakes with significant seasonal dynamics and mixed land–water elements within the observed pixels.

To further validate our findings and the universality of the W-shaped $T_b$ series beyond Lake Ulansu, we recently processed four additional lakes that exhibited similar seasonal characteristics (Fig. R2). These lakes, such as Lake Ulansu, feature mixed pixels where the $T_b$ series is influenced by both water bodies and surrounding land. These additional studies underscore that W-shaped $T_b$ series are indeed common in seasonally freezing lakes characterized by mixed pixels. Moving forward, we plan to enhance the DMTT algorithm further, aiming to expand its utility to a broader range of small- and medium-sized lakes globally. This advancement will provide deeper insights into the ice phenology of lakes under the influence of

climate change, particularly for periods and regions where high-resolution optical satellite data are not available. Figure R2 has been added to Appendix F of our manuscript, and future research directions are detailed on lines 472-475.

[Figure]

**Figure R2: (a-d) Landsat 8 images of Lake A, Lake B, Lake C, and Lake D, areas within red rectangles where mixed brightness temperature pixels were analyzed. (e) Brightness temperature time series from August 1, 2019, to August 1, 2020, for each lake.**

22. Line 190: I would add more information here about how your algorithm is different from the MTT since I feel like this is where the novelty of the study lies. Much of that information is in the appendices, and I would move that information into the main manuscript.

Reply: We have relocated the detailed comparison, previously found in the appendices, into the main body of the manuscript (Section 4.1 Algorithm evaluation). This adjustment ensures that the enhancements introduced by the DMTT are immediately apparent.

23. Line 193: How complex are the mixed pixels? The five that you show were mostly water and not near the edge. Do you still expect these pixels to be mixed?

Reply: Despite the selection of five pixels with a high proportion of water area, these pixels are still considered mixed because there are no pure water pixels within Lake Ulansu. The primary reason for selecting pixels with larger water areas is to minimize the influence of the land surface brightness temperature ($T_b$) and achieve more accurate ice phenology results for Lake Ulansu. These five pixels are located primarily in the central area of the lake where the water depth is relatively consistent, thus reducing the effects of uneven ice formation due to depth variation.

In previous studies that utilized satellite data to analyze lakes, a buffer zone was often established along the lake edge to avoid pixel contamination and ensure the quality of pure pixels (Cai et al., 2022; Giroux-Bougard et al., 2023; Heinilä et al., 2021; Korver et al., 2024; Kuluwan et al., 2023). However, Lake Ulansu has an elongated shoreline, and despite some pixels having a high proportion of water area, all pixels within the lake are subject to contamination by aquatic vegetation or land surfaces.

In this study, we used Calibrated Enhanced-Resolution Passive Microwave Daily EASE-Grid 2.0 Brightness Temperature (CETB) data at 37 GHz H-polarization with a high spatial resolution of 3.125 km. The CETB data were obtained from various satellite sensors with coarse spatial resolutions (~25 km), including SMMR on Nimbus 7, SSM/I and SSMIS on the DMSP satellite series, and AMSR-E on Aqua. Although the CETB data boast a higher spatial resolution than the original coarse-resolution pixels do, they are still derived from coarse-resolution data via the radiometer version of the scatterometer image reconstruction (rSIR) algorithm. This enhancement results in $T_b$ data with resolutions ranging from 3.125 km to 12.5 km. Therefore, even with enhanced resolution, the pixels remain mixed to some extent due to the averaging and enhancement process (Section 3.1.1).

24. Line 199: How does including rules like "above 0 degrees" and "below 0 degrees" make mixed pixel classification better? I think I understand your logic, but I would like it spelled out in the manuscript

Reply: The decision to utilize specific temperature thresholds of "above 0 degrees" and "below 0 degrees" in our analysis is crucial for enhancing the accuracy of mixed pixel classification, particularly in the complex environment of Lake Ulansu. These thresholds are based on the fundamental physical properties of water and ice, where 0 °C marks the freezing point under atmospheric pressure, serving as a natural demarcation between the liquid and solid-states. This becomes particularly important in the context of Lake Ulansu, where mixed pixels often exhibit numerous abrupt change points (ACPs) in the brightness temperature ($T_b$) time series.

By implementing a rule that transitions from water to ice are recognized only at air temperatures below freezing point, we effectively reduce misclassification that might otherwise occur from transient $T_b$ fluctuations or the influence of nonwater elements within the pixel, such as vegetation. Conversely, the rule that transitions from ice to water only occurs at air temperatures above freezing point helps accurately identify melting events, which is crucial for avoiding false interpretations of thawing during cold conditions. These air temperature-based rules simplify the classification process, providing clear and objective criteria that enhance our DMTT algorithm's ability to accurately calculate ice phenology in mixed pixels.

Revised text in the manuscript "*In addition to adhering to the ACP criteria outlined by Du et al. (2017), we tailor the DMTT algorithm to detect ACPs by accounting for seasonal variations and specific thermal conditions. We conducted separate ACP detection for $T_b$ from August to December and from January to July. To ensure the accuracy of the ACP in indicating freezing transitions, ACPs detected from August to December were validated only if they were accompanied by air temperatures below the freezing point. Similarly, for ACPs from January to July, a prerequisite condition of air temperatures above the freezing point was applied to confirm melting transitions. The purpose of these two enhancements was to accommodate the variations in mixed pixels during the transitions between water and ice.*"

25. Line 201: what does the bar symbol above Tb1 mean? What does b1 mean and b2 mean?

Reply: In the process outlined, $\overline{T_{b1}}$ and $\overline{T_{b2}}$ represent the mean brightness temperature ($T_b$) calculated 20 days before and after each detected abrupt change point (ACP), respectively. These averages are crucial for accurately determining the transitions between different states of water and ice.

The notation $\overline{T_{b1}}$ specifically indicates the mean $T_b$ calculated from the time series data before the ACP, providing a baseline against which changes due to freezing can be detected. Conversely, $\overline{T_{b2}}$ represents the mean $T_b$ after the ACP, which helps in identifying thawing events. By averaging the $T_b$ values around each ACP, we minimize the influence of short-term fluctuations in $T_b$.

The terms b1 and b2 mentioned in the algorithm pertain to the calculated thresholds for freezing and melting, respectively. The freezing threshold is derived as the mean of the minimum $\overline{T_{b1}}$ and its corresponding $\overline{T_{b2}}$ within the same group of ACPs, indicating the transition from water to ice. The melting threshold, on the other hand, is calculated as the mean of the minimum $\overline{T_{b2}}$ and its associated $\overline{T_{b1}}$, facilitating the accurate detection of ice-to-water transitions.

26. Line 202: If means the average over 20 days, then how can there be a minimum?

Reply: The term "minimum" in our analysis refers not to the minimum of the brightness temperature ($T_b$) within those 20 days but to the selection of the lowest $\overline{T_{b1}}$ and $\overline{T_{b2}}$ values from different groups of ACPs for further calculations. Essentially, after computing the averages for each 20-day period surrounding each ACP, we then examine these average values across multiple ACPs to identify the lowest averages: where $\overline{T_{b1}}$ for freezing transitions and $\overline{T_{b2}}$ for thawing transitions.

This approach allows us to determine the most representative thresholds for freezing and melting by selecting the most stable and consistent average values, which reflect the most significant changes in the ice/water state. By using the minimum of these averages, we ensure that our thresholds for ice formation and melting are based on the most pronounced and reliable data points, thereby enhancing the accuracy of the DMTT algorithm in determining ice phenology.

27. Line 214: Were the multisource Sentinel, Landsat, and MODIS?

Reply: Thank you for highlighting the need for clarity regarding the multisource optical satellite data used to validate the DMTT algorithm.

Revised text in manuscript "*The results of the DMTT algorithm were validated through a comparison with ice phenology data obtained from multisource optical satellite data from 2000 to 2023 (Section 3.1.2).*"

28. Lines 215 -220: I think some of this text belongs in the results section to test the validity of the DMTT model

Reply: We have relocated the relevant text, which previously detailed the validation of the DMTT algorithm, from its initial location to the results section (Section 4.1).

29. Line 235: which phenology datasets did you use? Which meteorological datasets did you use?

Reply: The text in the manuscript "*To train and validate the RF model, we randomly divided the dataset, which includes ERA5 meteorological factors and ice phenology derived from $T_b$ data for the period 1979 to 2023, into two subsets.*"

30. Line 255: Did you validate or ground truth your model using in-situ data?

Reply: In our study, we employed a robust validation approach. To ensure comprehensive coverage and unbiased validation across more than four decades, we randomly divided this dataset into two subsets: 70% was used for training the model, and 30% was used for validation. This random stratification

prevents any temporal biases that might arise from the use of distinct periods for training and validation (e.g., 1979–1990 for validation and 1991–2023 for training). Furthermore, for additional validation of the meteorological data, refer to Appendix A, where the performance of the ERA5 data is assessed. The ice phenology estimates derived from the brightness temperature calculations were cross-validated against corresponding results obtained from optical satellite data (Section 4.1).

31. Figure 6: How did you choose 1982 as the threshold change? I would include significance values in the figure directly. For example, only drawing lines and indicating trends if the results are significant

Reply: We chose 1982 as the threshold change based on the analysis of the 21-year moving averages (10 years before and after) of the freeze-up start (FUS), break-up end (BUE), and ice cover duration (ICD) periods. As shown in Fig. R3 (included in Appendix D), after 1982, there was a noticeable trend where FUS was delayed and the ICD was shortened. This marked change in trends approximately 1982 led us to select it as the threshold point.

To provide a more detailed depiction of this reversal, we divided the whole period into two subperiods (1941–1982 and 1983–2023) for further analysis. The corresponding statistical results are presented in Table 2 of the manuscript. Additionally, we have now included significance values ($p$ values) directly in Figure 6 to indicate the statistical significance of the trends.

Revised text in the manuscript "*The results of the DMTT algorithm were validated through a comparison with ice phenology data obtained from multisource optical satellite data from 2000 to 2023 (Section 3.1.2). Figure 4 reveals a remarkably high correlation for the FUS dates (r = 0.92), with a minimal MAE of 2.00 days and an RMSE of 2.56 days. The correlation for the BUE date is slightly lower (r = 0.87), with a somewhat greater MAE of 2.67 days and an RMSE of 3.25 days. However, the MAEs and RMSEs for the ICDs calculated based on the difference between the BUE and FUS dates remain within 5 days, indicating no systematic bias but random errors.*"

[Figure]

**Figure R3: 21-year moving averages of ice phenology for Lake Ulansu. (a): Freeze-up start dates. (b): Break-up end dates. (c): Ice cover duration. The pink vertical line indicates the year 1983.**

[Figure]

**Figure 6: Ice phenology results for Lake Ulansu from 1941 to 2023. (a) Freeze-up start with trend lines. (b) Freeze-up start date distribution. (c) Break-up end with trend lines. (d) Break-up end date distribution. (e) Ice cover duration with trend lines. (f) Ice cover duration distribution.**

32. Line 284: Do you have a citation for the statement "ice phenology is primarily determined by local meteorological conditions"?

Reply: We have added citations to support the statement that ice phenology is primarily determined by local meteorological conditions. See details on lines 291-292.

33. Line 292: This comment about Lake Mendota feels more like a methods comment

Reply: The text in the manuscript "*The observed air temperature fluctuations echo broader climate changes and show similar trends to those observed in other lakes (Magee et al., 2016; Newton and Mullan., 2021).*"

34. Figure 7: I would recommend moving this to the supplementary text, I feel that the preceding text does a good job summarizing the information.

Reply: Revised accordingly.

35. Line 331: Is there evidence that the preceding 12 months weather data could impact ice formation? I am not sure that each monthly correlation is necessary.

Reply: Thank you for your insightful question. We re-evaluated the monthly correlations and decided to focus on the months from September to March, which are more directly relevant to ice phenology and correspond to the months used in our random forest model calculations for ice phenology. We also modified the range of results in Figure 7.

[Figure]

Figure 7: Correlation coefficients between meteorological factors and ice phenology in different months. Significance

levels for correlation coefficients are denoted by * ($p < 0.05$) and ** ($p < 0.01$).

36. Line 330: I am not sure that I followed the logic regarding solar radiation and algae and attenuation. I was wondering if the authors would consider expanding on this mechanism

Reply: In eutrophic lakes, due to the high concentrations of algae and suspended particles, the absorption and penetration of solar radiation differ significantly from those in clear lakes. Eutrophic lakes are rich in nutrients (such as nitrogen and phosphorus), promoting the proliferation of algae and phytoplankton (Li et al., 2024). These algae and suspended particles absorb and scatter solar radiation, especially blue light (450–495 nm). After being absorbed, this light energy is converted into heat or used for photosynthesis (Lin et al., 2024).

Because blue light is strongly absorbed, the penetration depth of light in water is significantly reduced. In clear lakes, light can penetrate deeper into the water column, but in eutrophic lakes, light is mainly absorbed and scattered by algae and suspended particles near the surface, reducing the penetration depth of light. Consequently, the heating effect of solar radiation on water is diminished.

The revised text in the manuscript "*For Lake Ulansu, the correlation may be due to the high attenuation coefficient caused by the presence of numerous suspended particles and algae, which are common in eutrophic lakes (Yang et al., 2020). These particles and algae absorb and scatter solar radiation, especially blue light (450–495 nm), reducing the penetration depth of light in water (Lin et al., 2024). As a result, the heating effect of solar radiation is confined primarily to the surface layers, leading to a diminished overall heating effect on the water.*"

37. Figure 8: Are you correlating FUS and BUE and ICD with months that occur after ice cover? It seems that in BUE and ICD the months stop at March, which feels appropriate. But, for freeze-up start, the months from Dec to Mar occur after the event and cannot correlate with the event as they cannot influence the event anymore.

Reply: We have revised Figure 7 to focus on the relevant months for each ice phenology event. We appreciate your attention to this detail, which enhances the accuracy and relevance of our analysis.

38. Section 5, Discussion: I would recommend starting this section with an overarching paragraph of main findings before delving into each subsection

Reply: We added this overarching paragraph to the Discussion, which reads "*The novel DMTT algorithm developed in this study has proven essential for accurately discerning ice and water states in a lake with*

*mixed pixel challenges. In this section, we discuss the uncertainties associated with methodologies, compare our findings with those of other studies, and elaborate on the implications of the results for understanding the impacts of climate change on lake ice phenology.*"

39. Line 274: If the pixels chosen in the CETB database are such that they reduce the effects of mixed pixels, how do you know if your DMTT algorithm does a better job at overcoming mixed pixels.

Reply: The effectiveness of the DMTT algorithm in managing the challenges posed by mixed pixels in Lake Ulansu, even with the selection of higher resolution CETB data at 3.125 km. While these selected pixels indeed have greater water coverage, reducing the influence of nonwater elements, they do not entirely eliminate the complexities associated with mixed pixel conditions. Compared with the MTT algorithm, the DMTT algorithm offers significant improvements in addressing these complexities (Section 4.1). By employing a double-threshold approach, the DMTT algorithm specifically adapts to the nuanced variations between ice and water states observed in Lake Ulansu. This method allows for a more accurate identification of state changes in mixed pixels, reducing the likelihood of undetected transitions, which are more common with simpler, single-threshold approaches. Furthermore, our comparative analysis between the DMTT and MTT algorithms demonstrates that the DMTT provides a better fit for the specific conditions of Lake Ulansu (Section 4.1). It more effectively reduces the incidence of undetected states, enhancing the dataset's overall accuracy. This is supported by the robust validation against multisource optical satellite data, which corroborates the DMTT algorithm's performance in handling complex mixed pixel scenarios.

40. Line 298: I would elaborate on this further. If most lakes in the Northern Hemisphere that have data are deep lakes, but shallow lakes are abundant? Then what about the comparison between the two makes it interesting? (I agree with you that it would be interesting, but I would like to know your reasoning)

Reply: The comparison between shallow and deep lakes is indeed intriguing and scientifically valuable for several reasons:

(1) Methodological insights: Comparing different lake types can also refine our methodological approaches in limnology and cryosphere sciences. It challenges existing algorithms and encourages the development of new techniques suitable for diverse conditions, as demonstrated by our DMTT algorithm for Lake Ulansu.

(2) Differential thermal dynamics: Shallow lakes, such as Lake Ulansu, have minimal water volumes and shallow depths which lead to faster thermal responses. This characteristic makes them highly sensitive

to atmospheric temperature changes, allowing them to serve as early indicators of climate change effects. In contrast, deeper lakes with larger volumes exhibit greater thermal inertia, delaying their response to climatic shifts. By comparing these systems, we can better understand the spectrum of lake responses to global warming, providing a broader perspective on thermal dynamics under changing climatic conditions.

(3) Ice cover variability: The dynamics of ice cover in shallow lakes are markedly different from those in deeper lakes because of their rapid thermal response and shallower thermocline. Shallow lakes tend to freeze and thaw more quickly, which could lead to a shorter ice cover duration (ICD) under warming scenarios. In deeper lakes, the ICD is generally longer because of its their ability to retain cold temperatures. Studying these differences helps illuminate how variations in lake depth influence ice phenology, which is crucial for predicting ecological impacts across different lake types.

These points have been elaborated upon in the revised manuscript to better articulate the scientific significance of comparing shallow and deep lakes, thereby clarifying the unique contributions of our study to the fields of limnology and climate science. See details on lines 404-442.

41. Line 402: Can you qualify why you chose to compare Lake Ulansu with Qinghai Lake?

Reply: This comparison is instrumental in highlighting the distinct phenological responses of lakes with different physical and ecological characteristics under similar climatic latitudes. Qinghai Lake, one of the largest saline lakes in a high-altitude environment and a widely studied subject, offers a robust dataset, including long-term ice phenology data. These high-quality, extensive datasets make Qinghai Lake an ideal reference point for assessing trends and methodologies, enhancing the validity of our comparative analysis.

The key differences in their physical attributes—such as depth and salinity—result in varied responses to climatic factors, particularly air temperature and solar radiation, which influence their freeze-up start and break-up end dates. By comparing these two lakes, we aim to illustrate how varying lake characteristics can affect ice phenology, thus providing broader insights into how different types of lakes might respond to climate change. Qinghai Lake, with its larger volume and more saline water, has different thermal inertia and sensitivity to temperature changes than does the more responsive Lake Ulansu. This comparative analysis not only enriches our understanding of lake response but also supports the development of more nuanced models for predicting lake ice dynamics across diverse ecological settings. See details on lines 407-420.

42. 5.2 Comparison with other lakes: In general, I think this section needs to be better qualified, why did you choose the comparisons with Qinghai Lake or the northern European lakes? What are some major consequences of the similarities/differences? How do you think adding shallow lakes will change our contextual understanding of lake ice, globally? Or locally?

Reply: We have restructured this section to provide a clearer rationale for our choice of comparative studies with Qinghai Lake and the lakes in Northern Europe. This revision includes a more detailed discussion on why these particular comparisons are pertinent and how they contribute to the broader understanding of lake ice phenology under varying climatic and geographic conditions.

43. 5.3 Conclusion: This section is extremely well written and summarizes the study well

Reply: Thank you for your encouraging feedback on the Conclusion section of the manuscript.

44. Line 455: Is there a citation associated with this, or is this in comparison to the other lakes in section 5.3?

Reply: We have revised the statement to emphasize the pronounced reduction in ice cover duration (ICD) observed at Lake Ulansu over the last four decades. We note that this change is significantly more marked than the gradual trends in the ICD across other Northern Hemisphere lakes. See details on lines 455-459.

45. Line 54 add CERP acronym here

Reply: Revised accordingly.

[revised manuscript text omitted]

---

## Author Comment (AC2)

Reply to RC2

Note: the comments and authors' replies are in black and blue font, respectively. All changes in the revised manuscript are highlighted in yellow.

Lake ice phenology is a sensitive indictor to climate change.

The paper proposed a new double-threshold moving t test (DMTT) algorithm applied to CETB dataset, and reconstructed the ice phenology of Lake Ulansu from 1941 to 2023 using Random Forest based on the ERA5 data, which have been proved relatively good performance. The method is novel in identifying lake ice phenology and the influencing factors, but there still need some improvement.

(1) The introduction is not well organized, and need to improve.

(2) The logic of methodology is not clear. The overview of Data makes the reader confusing.

(3) The caption of figures should be concise, and the author need to check and improve them.

(4) The correlation between solar radiation and lake ice phenology is lower than we expected. The authors analyzed this in terms of month. Freeze-up and break-up date are one value in a winter, and other parameters varies each month, how to connect them together?

In conclusion, the subject of the study is suitable for "The Cryosphere" and can potentially be of interest for the journal's wide audience, and we suggest a major revision.

Reply: Thank you for your constructive comments on our manuscript, which are highly valuable to us during the revision. We have made major revisions to the manuscript accordingly, and the main points are listed as follows:

(1) We have restructured the Introduction to clarify the importance of studying shallow, vegetated lakes such as Lake Ulansu for ice phenology research. The revised sections now better highlight the novelty of our study and the progress of current methodologies and clearly state our research objectives. These changes can be found on lines 23-75 of the revised manuscript.

(2) We have revised Sections 3.1 and 3.2 to provide a clearer overview of the data sources and their specific uses within our study, as well as detailed descriptions of the methodologies employed. Figure 2 has been updated to visualize represent the flow from the raw data to the ice phenology results more clearly. Additional comparisons between our algorithm and existing methods, including MTT and field observations, are detailed in a new subsection, 4.1 Algorithm evaluation, to underline the enhancements our approach offers.

(3) The figure captions have been revised to be more concise and clear, enhancing the readability and

directness of the presented data. Specific changes have been made to Figures 1, 2, 3, 5, 6, B1, and D1.

(4) We elaborate on the unique characteristics of Lake Ulansu as a eutrophic lake. The presence of algae and suspended particles in Lake Ulansu can absorb and scatter solar radiation, mitigating the effects of warming on the water body. These details have been added in lines 344-349. Furthermore, we adjusted our analysis to focus on the relationships between meteorological factors from September to March and ice phenology, as illustrated in Figure 7.

[Figure]

**Figure 7: Correlation coefficients between meteorological factors and ice phenology in different months. Significance levels for correlation coefficients are denoted by * ($p < 0.05$) and ** ($p < 0.01$).**

Specific comments:

1. The Introduction is not clear and attractive. The meaning of long-term tracking of climate records such as lake ice phenology, is not well expressed. The research meanings in the first paragraph of Introduction are not closely related with the content, and the author should rewrite. The scientific problem and study goals are note clear.

Reply: We have thoroughly revised the first paragraph to better articulate the significance of long-term lake ice phenology tracking in the context of climate change and its ecological implications. We have also clarified the scientific problems and objectives of this study by emphasizing how the research fills critical

knowledge gaps in understanding complex lake systems under changing climatic conditions. The Introduction now effectively sets the stage for the methods and findings presented, underscoring the importance of ice phenology data for enhancing climate models and weather forecasting. See details on lines 23-35.

2. Line 38 the best quality is not appropriate.

Reply: The revised text in the manuscript "*In situ observations provide a wealth of information on ice phenology (Benson et al., 2011; Yang et al., 2020).*"

3. Line 41 MODIS just mentioned 8-day product, and the author should update recent work.

Reply: The text in the manuscript has been revised: "*For example, the MODIS Terra and Aqua products, specifically MOD10A1 and MYD10A1, have been utilized to determine lake ice phenology across the Tibetan Plateau from 2001 to 2017 (Cai et al., 2019).*"

4. Line 46 The active microwave is not suitable for the large lakes. This sentence should be improved, and supplement more contents.

Reply: We added this information to the Introduction: "*Active microwave remote sensing is utilized primarily for extracting ice phenology data from large water bodies because of its relatively low (> 10 days) temporal resolution (Antonova et al., 2016; Howell et al., 2009).*"

5. P47 One advantage of passive microwave is frequent revisit, which is not mentioned.

Reply: The original text has been modified to "*Passive microwave remote sensing provides data with long temporal coverage and frequent revisit times, although it offers coarse spatial resolution.*"

6. The second paragraph is too long, and should be divided. The core of this work belongs to passive microwave, and could be discussed alone.

Reply: Revised accordingly.

7. The fitting methods of lake ice phenology based on climate records (such as air temperature) are note well discussed, just listed some previous works.

Reply: We have revised the relevant text to better illustrate the practicality of our methodology. The updated section now emphasizes the importance of integrating diverse meteorological factors with ice

phenology data to increase the effectiveness of machine-learning models. See details on lines 59-66.

8. The author explained how the study organized, and the goals of this study is not clear. The last paragraph should be rewritten.

Reply: To clarify and specify, we have revised the last paragraph of the Introduction to better articulate these specifics: "*Specifically, our research strategy was composed of the following steps: (1) A new algorithm was developed to classify the ice and water states on the lake surface in brightness temperature data from the SMMR and SSM/I-SSMIS sensors for the period 1979–2023. (2) A random forest model was trained using the results in step (1) to reconstruct the ice phenology from 1941 to 1978. (3) The meteorological impact on the ice phenology of Lake Ulansu from 1941 to 2023 was analyzed to explore the key drivers of its variations.*"

9. The general description of ice regime should be added in Study are, like ice thickness, regular winter recreation.

Reply: We have amended the manuscript to include a detailed description of the ice thickness and winter recreation. See details on lines 91-92.

10. Line 75 300 need to check, or provide the citation.

Reply: The original text has been modified to "*The lake covers an area of 306 km$^2$.*"

11. Line 80 how to exclude the influence of vegetation?

Reply: Given the lake's narrow and irregular shoreline, we implemented several strategies to ensure the accuracy and reliability of our phenology assessments, which are detailed in Section 3.2 of the manuscript.

(1) To address the complex mixed pixels in Lake Ulansu, we developed and applied the DMTT algorithm. We adapted the detection of abrupt change points (ACPs) in the Tb time series by considering seasonal variations and air temperature constraints, ensuring that freezing and melting transitions are accurately identified even in the presence of mixed pixels. The algorithm calculates distinct thresholds for freezing and melting based on the mean $T_b$ values before and after each detected ACP, tailored specifically to accommodate the variability introduced by mixed pixels.

(2) We selected five $T_b$ grids with a water proportion greater than 0.70 to represent the lake surface status. This threshold selection is crucial for reducing the influence of mixed pixels and is specifically tailored to the unique geographical characteristics of Lake Ulansu.

These details are further elaborated in Section 3.2.1, providing a comprehensive view of how we address the challenges posed by mixed pixels at Lake Ulansu.

12. Line 83-84 remove the date.

Reply: Revised accordingly.

13. Line 85 The definition of hydrological year is not accurate. For example, HY 2022 lasted from August in 2021 to July in 2022.

Reply: To clarify and specify, we have revised the sentence to better articulate these specifics: "*In this study, each hydrological year is defined as beginning on August 1st and ending on July 31st of the following year. For example, HY2022 spans August 1, 2021, to July 31, 2022.*"

14. Line 90 The title of figure 1need to revise.

Reply: The original text has been modified to "*Figure 1: (a) Geographical context and elevation profile of Lake Ulansu within the Hetao Basin, Inner Mongolia. (b) CETB data grids with shaded areas representing brightness temperature pixels selected for Lake Ulansu surface identification. (c) Photographic depiction of aquatic reeds within Lake Ulansu. (d) On-ice instrumentation for field observations (Cao et al., 2021).*"

15. Line 95-105 The logic of flowchart is not clear, and Figure 2 should be revised. The input and output is not clear. The different colors have certain meanings? The method is too short.

Reply: We have thoroughly revised the flowchart to better illustrate the logical progression from data inputs through methodologies to outputs. In addition to revising the Fig. 2, we have also enhanced the corresponding textual descriptions within the manuscript. See details on lines 102-109.

[Figure]

**Figure 2: Reconstruction of the ice phenology of Lake Ulansu (1941–2023) based on ERA5, CETB, and optical satellite data via the double-threshold moving *t* test algorithm and random forest model.**

16. Line 119 high spatial resolution Tb data are essential, gramma error

Reply: Revised accordingly.

17. Line 124 These data were sourced from the SMMR on the Nimbus 7 satellite; the SSM/I on the F08, F10, F11, F13, F14 and F16 satellites; and the SSMIS on the F16 satellite. Move this to the above content.

Reply: We have carefully revised this section to clarify the selection and usage of CETB data for our study. See details on lines 122-123.

18. Figure 3 have two choices: (1) make the content short and concise, and add the figure in the supplementary materials; (2) move this part to Results

Reply: We agree that the best approach is to make the content short and concise and to move the detailed validation section, along with Figure 3, to Appendix A. This adjustment will streamline the main manuscript while still providing access to the detailed validation results for interested readers. We believe that this modification maintains the narrative flow of the paper and allows us to focus more directly on the results in the main text.

19. Figure 4 "between 1 August and 31 July from 1979 to 2023 for Lake Ulansu": check this. "The solid lines represent the interannual average brightness temperature and air temperature time series, respectively. " delete this sentence. Lack the description of Figure 1 (a), just (b) appears."

Reply: We appreciate your attention to detail and have made the following revisions to address your concerns:

(1) We verified the date range "between 1 August and 31 July from 1979 to 2023" for Lake Ulansu and confirmed its accuracy. To clarify, each series represents data from a hydrological year, defined as 1 August to 31 July of the following year.

(2) We have removed the sentence "The solid lines represent the interannual average brightness temperature and air temperature time series, respectively." to streamline the figure caption."

(3) We have added a description for part (a) of the figure.

Revised Figure 3 Caption:

"*(a) Time series of brightness temperatures (blue dashed lines) for each hydrological year from 1979 to 2023. A hydrological year (HY) was defined from 1 August to 31 July of the following year for Lake Ulansu. (b) The brightness temperature (blue solid line) and air temperature (red solid line) for HY2001.*

*The ice and water statuses were determined via the double-threshold moving t test algorithm, where the red and blue shaded regions represent the water and ice states, respectively.*"

20. Line 174 "proportion lake water greater than 0.70" how the thresholds are determined? Why do not use 0.8/0.2?

Reply: In our study, we opted for a threshold of 0.70 for the proportion of lake water in the $T_b$ grids to ensure a significant representation of water surfaces while still capturing sufficient spatial coverage across Lake Ulansu. The choice of this threshold was driven by the need to balance the inclusion of enough grid cells for robust statistical analysis and minimize the influence of adjacent land and vegetation within each pixel.

Reference thresholds of 0.80 or 0.20, which might be familiar from studies utilizing optical satellite data such as MODIS for ice phenology, were considered. However, these thresholds typically cater to different analysis objectives and sensor characteristics. For passive microwave data, especially in the context of lakes with complex shorelines such as Lake Ulansu, a threshold of 0.70 provides a pragmatic compromise between data availability and the accuracy of representing lake surface conditions.

21. Line 236 "Seventy percent" and "30 %", keep the same expression.

Reply: Revised accordingly.

22. Line 258 ERA5 have various types of climate data, which one you used? It is not clear. The input and output of RF is confusing.

Reply: We have clarified the description of the ERA5 meteorological data used as input for our random forest model from 1941 to 1978. The revised sentence now reads: "*After the RF model was established and validated, we used ERA5 meteorological data from 1941 to 1978 as input features to obtain the historical ice phenology. These data included average air temperature, wind speed, solar radiation, and cumulative precipitation for the months of September to November and January to March each year.*" We have also revised Figure 2 to more clearly illustrate the input and output processes within our modeling workflow.

23. Figure 6 Please add the math equation and basic index of linear regression. Please add the lengend of different lines, it would be better to remove the description of dashed lines. (a)-(f) explained separately.

Reply: Thank you for your valuable suggestions regarding Figure 6. We have revised the figure to include

the mathematical equations and basic indices of the linear regression for each segment. We have also updated the legend to clearly differentiate between the various lines depicted in the graphs and have removed the description of the dashed lines from the legend for clarity. The explanations for parts (a) through (f) have been distinctly outlined to ensure that each component is clearly understood.

[Figure]

**Figure 6: Ice phenology results for Lake Ulansu from 1941 to 2023. (a) Freeze-up start with trend lines. (b) Freeze-up start date distribution. (c) Break-up end with trend lines. (d) Break-up end date distribution. (e) Ice cover duration with trend lines. (f) Ice cover duration distribution.**

24. Line 280 add the yearly changing rates.

Reply: We have updated the manuscript to include the yearly changing rates for ice phenology trends in Lake Ulansu. The revised text now reads: "*Overall, from 1941 to 2023, the ice phenology in Lake Ulansu exhibited several notable features. The FUS occurred between the 93rd and 119th days, with an average date of approximately the 107th day (November 15), showing a slight delaying trend of 0.07 d yr$^{-1}$. The BUE ranged from the 223rd to the 253rd day, typically occurring around the 237th day (March 25), with an advancing trend of 0.01 d yr$^{-1}$. The ICD spans from 115 to 154 days, with an average of approximately 130 days, decreasing by 0.08 d yr$^{-1}$ over the study period.*"
."

25. Line 282 The title of 4.2 is not proper.

Reply: We have revised the title of this section to "*Impact of meteorological factors on ice phenology*"

26. Table 3 Linear tread is yearly changing rates? Please add the unit. The linear tread of wind speed is 0. Check this?

Reply: Thank you for your attention to the details in Table 3. We have now added the appropriate units to indicate that the linear trends represent yearly changing rates, expressed in the respective units of each ERA5 parameter per year. With respect to the wind speed trend, our analysis reveals a minimal change, indeed nearing zero, which reflects the stable wind conditions over the study period at Lake Ulansu.

27. Figure 8 The correlation between solar radiation and lake ice phenology is lower than we expected.

Reply: In eutrophic lakes, due to the high concentrations of algae and suspended particles, the absorption and penetration of solar radiation differ significantly from those in clear lakes. Eutrophic lakes are rich in nutrients (such as nitrogen and phosphorus), promoting the proliferation of algae and phytoplankton (Li et al., 2024). These algae and suspended particles absorb and scatter solar radiation, especially blue light (450–495 nm). After being absorbed, this light energy is converted into heat or used for photosynthesis (Lin et al., 2024).

Because blue light is strongly absorbed, the penetration depth of light in water is significantly reduced. In clear lakes, light can penetrate deeper into the water column, but in eutrophic lakes, light is mainly absorbed and scattered by algae and suspended particles near the surface, reducing the penetration depth of light. Consequently, the heating effect of solar radiation on water is diminished.

The revised text in the manuscript "*For Lake Ulansu, the correlation may be due to the high attenuation coefficient caused by the presence of numerous suspended particles and algae, which are common in eutrophic lakes (Yang et al., 2020). These particles and algae absorb and scatter solar radiation, especially blue light (450–495 nm), reducing the penetration depth of light in water (Lin et al., 2024). As a result, the heating effect of solar radiation is confined primarily to the surface layers, leading to a diminished overall heating effect on the water.*"

28. As for random forest, we have following questions: (1) The abbreviation of correlation coefficients is r, not R. Need to check the whole manuscript. (2) The evaluate the performance of RF, the determination coefficients $R^2$ is more usually used, rather than R. (3) How the author avoid overfitting in the work? Need to explained more clearly. (4) Why the author chose random forest rather than other methods?

Reply: We have carefully reviewed and addressed each point as follows:

(1) We have revised the notation for correlation coefficients throughout the manuscript from "*R*" to "*r*" to align with standard statistical notation practices.

(2) We have updated the manuscript to use the determination coefficient "$R^2$" instead of "$r$" when discussing the performance of the random forest model.

(3) We determined that using 20 trees in the random forest model provided a balanced complexity that effectively captured the underlying patterns without overfitting. This decision was based on observing diminishing returns on model performance metrics with more than 20 trees (Appendix C). We implemented 3-fold cross-validation during the training phase to ensure that the model did not learn the noise and outliers of the training data, thereby enhancing its ability to generalize to unseen data. We continuously monitored the model's performance on both the training and validation sets to detect signs of overfitting, which was characterized by high performance on the training data but poor performance on the validation data.

(4) The random forest is robust in handling nonlinear relationships between variables, which is often the case in ecological and climatological studies. This method allows for an effective assessment of feature importance, which is crucial for understanding the influence of different meteorological factors on ice phenology. Compared with other models, random forest models are generally more resilient to overfitting, especially when the correct number of trees and other hyperparameters are chosen. Previous studies have successfully used random forest models in similar contexts (Anilkumar et al., 2023; Ruan et al., 2020), providing a tested framework for our analysis.

29. The two paragraphs in Author contribution are repeatedly expressed.

Reply: Revised accordingly.

**References:**

Anilkumar, R., Bharti, R., Chutia, D., and Aggarwal, S. P.: Modelling point mass balance for the glaciers of the Central European Alps using machine learning techniques, Cryosphere, 17, 2811–2828, doi:10.5194/tc-17-2811-2023, 2023.

Antonova, S., Duguay, C., Kääb, A., Heim, B., Langer, M., Westermann, S., and Boike, J.: Monitoring Bedfast Ice and Ice Phenology in Lakes of the Lena River Delta Using TerraSAR-X Backscatter and Coherence Time Series, Remote Sens., 8, doi:10.3390/rs8110903, 2016.

Benson, B. J., Magnuson, J. J., Jensen, O. P., Card, V. M., Hodgkins, G., Korhonen, J., Livingstone, D. M., Stewart, K. M., Weyhenmeyer, G. A., and Granin, N. G.: Extreme events, trends, and variability in Northern Hemisphere lake-ice phenology (1855–2005), Clim. Change, 112, 299–323, doi:10.1007/s10584-011-0212-8, 2011.

Cai, Y., Ke, C.-Q., Li, X., Zhang, G., Duan, Z., and Lee, H.: Variations of lake ice phenology on the Tibetan Plateau from 2001 to 2017 based on MODIS data. J. Geophys. Res.-Atmos., 124, doi:10.1029/2018JD028993, 2019.

Howell, S. E. L., Brown, L. C., Kang, K.-K., and Duguay, C. R.: Variability in ice phenology on Great Bear Lake and Great Slave Lake,

*Northwest Territories, Canada, from SeaWinds/QuikSCAT: 2000–2006, Remote Sens. Environ., 113, 816–834, doi:10.1016/j.rse.2008.12.007, 2009.*

*Li, H., Song, C., Yang, L., Qin, H., Cao, X., and Zhou, Y.: Phosphorus supply pathways and mechanisms in shallow lakes with different regime, Water Resour., 193, 116886, doi:10.1016/j.watres.2021.116886, 2021.*

*Lin, X., Wu, X., Chao, J., Ge, X., Tan, L., Liu, W., Sun, Z., and Hou, J.: Effects of combined ecological restoration measures on water quality and underwater light environment of Qingshan Lake, an urban eutrophic lake in China, Ecological Indicators, 163, doi:10.1016/j.ecolind.2024.112107, 2024.*

*Ruan, Y., Zhang, X., Xin, Q., Qiu, Y., and Sun, Y.: Prediction and Analysis of Lake Ice Phenology Dynamics Under Future Climate Scenarios Across the Inner Tibetan Plateau, J. Geophys. Res.-Atmos., 125, doi:10.1029/2020jd033082, 2020.*

*Yang, Q., Song, K., Hao, X., Wen, Z., Tan, Y., and Li, W.: Investigation of spatial and temporal variability of river ice phenology and thickness across Songhua River Basin, northeast China, Cryosphere, 14, 3581–3593, doi:10.5194/tc-14-3581-2020, 2020.*

*Yang, W., Xu, M., Li, R., Zhang, L., and Deng, Q.: Estimating the ecological water levels of shallow lakes: a case study in Tangxun Lake, China, Sci Rep, 10, doi:10.1038/s41598-020-62454-5, 2020.*